mathematical modelling/applied mathematics/ systems biology

Turing pattern, positional information, algebraic topology, algebraic biology, model distance, model equivalence

**Author for correspondence:**
Sean T. Vittadello
e-mail: sean.vittadello@unimelb.edu.au

# Model comparison via simplicial complexes and persistent homology

## Sean T. Vittadello and Michael P. H. Stumpf

School of BioSciences and School of Mathematics and Statistics, The University of Melbourne, Parkville, Victoria 3010, Australia

 STV, 0000-0002-4476-6732; MPHS, 0000-0002-3577-1222

In many scientific and technological contexts, we have only a poor understanding of the structure and details of appropriate mathematical models. We often, therefore, need to compare different models. With available data we can use formal statistical model selection to compare and contrast the ability of different mathematical models to describe such data. There is, however, a lack of rigorous methods to compare different models *a priori*. Here, we develop and illustrate two such approaches that allow us to compare model structures in a systematic way by representing models as simplicial complexes. Using well-developed concepts from simplicial algebraic topology, we define a distance between models based on their simplicial representations. Employing persistent homology with a flat filtration provides for alternative representations of the models as persistence intervals, which represent model structure, from which the model distances are also obtained. We then expand on this measure of model distance to study the concept of model equivalence to determine the conceptual similarity of models. We apply our methodology for model comparison to demonstrate an equivalence between a positional-information model and a Turing-pattern model from developmental biology, constituting a novel observation for two classes of models that were previously regarded as unrelated.

## 1. Introduction

Scientific models are representations of the physical world that isolate features of interest through various levels of abstraction [1]. The complexity of biological phenomena, exemplified in systems biology, necessitates the development of models to further the understanding of these systems [2–5].

There are many approaches to modelling biological systems, including continuum models [6–8], rule-based models [9–11], network models [12–14] and mechanistic models [5,15], which are not necessarily mutually exclusive classifications. While a model should represent the corresponding phenomenon as

faithfully as possible, it is also a requirement that the model is manageable. It is neither practicable to develop a complete model of a biological system, due to *combinatorial complexity* [3,10,16], nor is it necessary, since many components of a given system have relatively small effects on the system behaviour [4,17].

A biological system may be represented by various alternative models, which can differ with respect to the specific features isolated by each model and the interconnections between these features. Contrarily, similar models may represent ostensibly divergent biological systems, where the difference between the models may be only conceptual or, indeed, simply different parameterizations [18]. The ability to rigorously and efficiently compare models is therefore imperative for understanding complex biological systems, and can elucidate the relationship between apparently distinct phenomena [19–23]. Model comparison is, however, a non-trivial task for multiple reasons: the various representational forms of models; the different levels of abstraction and granularity between models; and the need for a systematic formalism that provides elucidation of conceptual similarities between models. In particular, we require a way of obtaining distances between models: with a meaningful distance between models we can start to cluster 'similar' models and look for shared characteristics; we can explore how 'different' proposed alternative models for a given process really are; and we can use them to distill design principles from large sets of models that have been proposed to model a given biological process or problem.

There are very few attempts to establish distances between models [24,25], and many of the candidates for a distance between models have obvious shortcomings: graph-based distances on model structures, for example, fail because many dynamical systems cannot be described in terms of simple graphs [18]; other well-defined distances used in functional analysis are too limited in scope, as are distances among e.g. stoichiometric matrices, which only apply for certain types of models; semantic distances between model formulations in e.g. CellML or SMBL, can depend too strongly on inconsistencies in modelling terminology, and also can fail to distinguish between structural and semantic differences; distances among model outputs such as information-theoretical distances, distances used in control theory, or correlation, (dis)similarity measures require assessment of model outputs and generally do not allow a direct structural comparison between models; finally, coding distances or the minimum description length would be promising candidates for distances between models. We could consider these, but they, too, have shortcomings, especially for different modelling frameworks (say ordinary differential equations or stochastic Petri-nets used to model the same scenario [26]).

Therefore, in this article, we present a novel formalism for model comparison that is flexible, systematic, quantitative and visual. Since models consist of a finite collection of components and their interconnections, we represent a given model as a combinatorial object where the components of the model are represented as labelled vertices. Furthermore, since the interconnections between model components may have dimension higher than the dyadic connections in combinatorial graphs, we employ simplicial complexes to represent the models which allow for interconnections of any finite dimension. Representing models as simplicial complexes ensures that any model can be represented within our framework, irrespective of form, granularity, and level of abstraction. We have to define model components—for this we may draw on domain expertise relevant to the scientific modelling problem at hand—and model comparison is then performed by comparison of the simplicial representations of the models given these components. Our framework is both flexible, incorporating different models and modelling approaches, and rigorous.

Given the model components, the simplicial representations provide a graphical form for the models; however, we can also obtain visual representations of the models as collections of persistence intervals by applying persistent homology to the simplicial complexes. Persistent homology identifies the global topological structure of a space that persists across multiple scales [27,28], and is often employed in topological data analysis to study high-dimensional datasets [29]. Persistent homology is robust with respect to small perturbations in the input data, and represents the qualitative structure of the input data in a compact manner [29]. While we use tools that are also employed in topological data analysis, our aims here are different: we are trying to represent models in a framework that allows for meaningful comparisons, rather than looking for topological structure in high-dimensional data. Our application of persistent homology is novel since our simplicial complexes correspond to models rather than data sets, and we use the persistence intervals associated with a simplicial complex as an alternative representation of the model and hence the corresponding biological system. The persistence-interval representations tend to facilitate the visual comparison between various models, which may be more difficult with the simplicial representations and particularly with the models in their original forms. Indeed, since three- and higher-dimensional simplices cannot be drawn in two

dimensions, only projected, representations as persistence barcodes provide immediate information for all dimensions in an often clearer visual form.

We provide two general methodologies for model comparison. The first is comparison by distance, which measures the difference between models in terms of the differences between the corresponding labelled simplicial representations, and can be calculated either directly using the simplicial complexes or indirectly with the corresponding persistence intervals. The second is comparison by equivalence, where we identify any equivalences between the components of the simplicial representations of different models, and then employ operations on the complexes to transform one into the other. Where this is possible we can learn about shared underlying characteristics between models.

As a particular application of our methodology we compare the two main classes of models for developmental-pattern formation, namely Turing-pattern and positional-information models. While there is an extensive literature for models of both Turing-pattern and positional-information systems, the relationship between these models, and between the corresponding biological systems, has long been unclear [30]. One main outcome of our model-comparison methodology is the demonstration that the Turing-pattern activator–inhibitor model is equivalent to the positional-information annihilation model from a significant conceptual perspective, where the fundamental difference between the two models is the location of the source of the gradient-forming morphogen. This novel observation suggests that the location of the morphogen source may influence the particular mechanism, namely Turing-pattern or positional information, for gradient formation.

The remainder of this article is organized as follows. In §2, we introduce our methodology for model comparison, which consists of five subsections: §2.1 provides a brief discussion of the required background in algebraic topology; in §2.2 we define our notion of a simplicial representation of a model; in §2.3, we develop the persistent homology of the simplicial representations of models; §2.4 provides the definition of distance between two simplicial representations of models; and, in §2.5, we establish our notion of model equivalence. We then apply our model-comparison methodology to two examples in §3, firstly a comparison of bisubstrate enzyme reactions, and secondly a comparison of Turing-pattern and positional-information models. Finally, in §4, we summarize the utility of our methodology for comparing models.

# 2. Methodology

## 2.1. Background

We begin with a brief, informal and self-contained discussion of the background in simplicial algebraic topology that is relevant for our work. A more detailed overview of simplicial complexes and homology is provided in the electronic supplementary material document. There is also a growing literature on related issues in topological data analysis [29], but there the aims and the details of implementation differ considerably from what we set out to achieve here.

A *p-simplex* is a generalization of a filled triangle in the plane to an arbitrary dimension $p$, whereby a point is a 0-simplex, a line segment is a 1-simplex, a filled triangle is a 2-simplex, a filled tetrahedron is a 3-simplex, and so forth. We can think of 0-simplices as vertices and 1-simplices as edges, so that a simple graph is therefore a set of 0-simplices and 1-simplices. A *k-face* of a simplex is a $k$-dimensional subsimplex, and the 0-faces of a simplex are said to *span* the simplex. A face $\tau$ of a simplex $\sigma$ is *proper* if $\tau \neq \sigma$.

A *simplicial complex* is a generalization of a simple graph, allowing for simplicies of dimension higher than one which represent higher-dimensional interactions. Specifically, a simplicial complex $K$ consists of a set of simplices such that: if $\sigma$ is a simplex in $K$ then every face of $\sigma$ is also in $K$; and, the nonempty intersection of any two simplices in $K$ is a simplex in $K$. A *simplicial subcomplex L* of a simplicial complex $K$ is a collection $L \subseteq K$ that is also a simplicial complex, and a *simplicial supercomplex M* of $K$ is a collection $M \supseteq K$ that is also a simplicial complex. The *k-skeleton* of a simplicial complex $K$ is the subcomplex $K^{(k)}$ consisting of the simplices in $K$ with dimension at most $k$. In particular, the 0-skeleton $K^{(0)}$ is the set of vertices and the 1-skeleton $K^{(1)}$ is the *underlying graph* of the simplicial complex $K$.

A simplicial complex $K$ is a topological space, formed from the union of its simplices, and simplicial homology characterizes the complex using algebraic techniques to compute the number of $k$-dimensional holes in the complex. So, for example, 0-dimensional holes are connected components and one-dimensional holes are non-bounding cycles of edges. The homology of $K$ depends on the underlying space of simplices and their intersections in $K$. Persistent homology studies the topological features,

namely the $k$-dimensional holes, of a complex $K$ across multiple scales, based on a *filtration* of a simplicial complex which is an increasing sequence of subcomplexes. Each $k$-dimensional hole in $K$ is created at some index in the filtration, and either persists through to the full complex $K$ or is annihilated at some intermediate index. Persistent homology therefore gives a multiset of *persistence intervals* $\mathcal{P}(K)$ describing the creation and annihilation of all $k$-dimensional holes in $K$. We denote by $\mathcal{P}_k(K)$ the submultiset of $\mathcal{P}(K)$ consisting of the intervals corresponding to $k$-dimensional holes. The persistence intervals can be visualized as a *persistence barcode*, displaying the intervals as horizontal line segments.

## 2.2. Simplicial representations of models

By a *model* we mean an abstraction of an observable phenomenon [1]. A specific detail of a given model is a *component* of the model, and all models consist of a finite number of components and their interconnections, which represent direct relationships between specific components. For example, a Turing-pattern system of reaction-diffusion equations, which is a model of a developmental-patterning process, includes components such as morphogens, diffusion, boundary conditions, reactions, and a morphogen gradient, along with the interconnections between particular components such as a morphogen and its diffusion.

In comparing models, including when defining a distance between models, it is important to ensure that model components are defined consistently across all models. Beyond this, the definition of model components typically arises naturally from the scientific-modelling context. It is essential to keep in mind that model comparisons are always with regard to the chosen level of conceptual detail used to represent the models. The intention of a simplicial representation of a model is to identify the concepts underlying the model, and their interconnections, thereby removing the formality of the mathematical representations to allow for the comparison of models within the same framework.

We begin by defining the set of model components:

**Definition 2.1 (Model components, generated model).** Let $\mathcal{C}$ be the set of all components that appear in the collection of models under consideration for comparison. We say that each such model is *generated* by a subset of components from $\mathcal{C}$.

**Notation 2.2.** Denote by $\mathbb{N}^*$ the set of positive integers, and by $\overline{\mathbb{N}^*}$ the set of extended positive integers $\mathbb{N}^* \cup \{+\infty\}$. For $n \in \mathbb{N}^*$, we denote the subset $\{m \in \mathbb{N}^* \mid m \leq n\}$ of $\mathbb{N}^*$ as $[n]$.

We represent model components as 0-simplices, and label the 0-simplices in a flexible manner to allow for more efficient or detailed labelling as required:

**Notation 2.3 (Representations of model components).** Let $\mathrm{Ord} : \mathcal{C} \to [|\mathcal{C}|]$ be a bijection that specifies an arbitrary order for the categorical data elements in $\mathcal{C}$. We represent each model component from $\mathcal{C}$ as a 0-simplex which is labelled variously as $i$, $v_i$ in recognition that 0-simplices are vertices, or the name of the model component given by $\mathrm{Ord}^{-1}(i)$, where $i$ is the position in the total order specified by Ord.

The ordering function Ord provides for computationally-efficient labelling of simplices. For example, a 2-simplex can be labelled as $\{1, 2, 3\}$, where the 0-simplices are $\{1\}$, $\{2\}$, and $\{3\}$, and the 1-simplices are $\{1, 2\}$, $\{1, 3\}$ and $\{2, 3\}$. Note that we have no need for the orientation of the simplicial complexes that is provided by the total ordering of the vertices by Ord, since we only consider simplicial homology over $\mathbb{Z}/2\mathbb{Z}$.

To a given model, we can associate various related simplicial complexes depending on the required levels of abstraction and component detail. Each associated simplicial complex consists of the relevant labelled 0-simplices that represent the model components, along with the one- and higher-dimensional simplices that represent the interconnections between the components, where the dyadic interactions are 1-simplices, the triadic interactions are 2-simplices, and so forth as required. The labelling of the 0-simplices induces a labelling of the higher-dimensional simplices through their spanning 0-simplices. Since the models of interest are associated with the same set of components $\mathcal{C}$, we can identify a particular labelled simplex, representing specific components and interconnections, within the different simplicial complexes associated with the models.

A general algorithm to associate a labelled simplicial complex $K$ with a model is as follows:

(i) Identify the agents in the model.
(ii) Identify the reactions and interactions of the agents in the model, which may provide for various levels of conceptual detail.

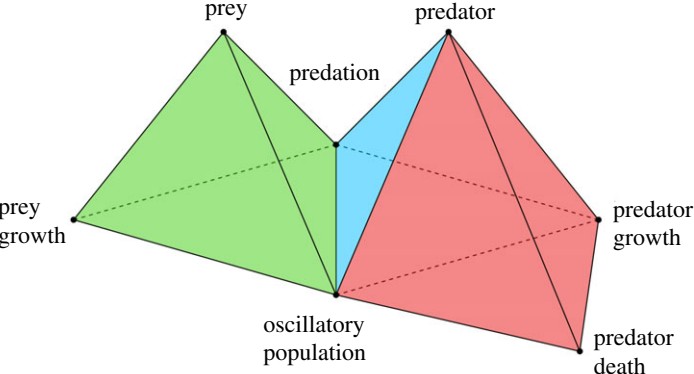

**Figure 1.** Simplicial representation of the Lotka–Volterra model. The representation contains higher-order simplices which capture the interactions between the components of the mathematical model. The vertices labelled 'prey', 'prey growth', 'predation' and 'oscillatory population' span a 3-simplex; the vertices labelled 'predator', 'predator growth', 'predation' and 'oscillatory population' span a 3-simplex; and the vertices labelled 'predator', 'predator growth', 'predator death' and 'oscillatory population' span a 3-simplex.

 (iii) Identify any boundary conditions for the model.
 (iv) Identify, if desired, the parameters in the model, which may allow for various levels of detail.
 (v) Define the set $\mathcal{C}$ as consisting of the components identified in steps (i)–(iv).
 (vi) The simplicial representation $K$ has $|\mathcal{C}|$ vertices with the labels from $\mathcal{C}$.
 (vii) The 1-simplices of $K$ correspond to direct connections between the relevant concepts, which are immediate from the model. To determine whether there should be a 1-simplex linking two vertices it may be easier to consider whether the absence of the 1-simplex is consistent with the model structure.
 (viii) Higher-dimensional simplices may or may not be required, depending on the particular model and the desired level of representation. Any higher-dimensional simplices must represent mutual interconnections between all model concepts involved.

With regard to steps (vii)–(viii) we note that, while all one- and higher-dimensional simplices of $K$ are completely determined by the model, such interconnections between the model components may not be obvious and may require a deep understanding of the model.

As an example we consider the Lotka–Volterra model ([31], ch. 3, p. 79), which is described by

$$\frac{\mathrm{d}x}{\mathrm{d}t} = x(a - by) \quad \text{and} \quad \frac{\mathrm{d}y}{\mathrm{d}t} = y(cx - d), \tag{2.1}$$

where $x$ is the prey density, $y$ is the predator density, and $a$, $b$, $c$ and $d$ are positive real constants. The underlying assumptions of the model are: in the absence of predation the prey population grows without bound; the reduction of the prey's *per capita* growth rate due to predation is proportional to both the prey and predator populations; in the absence of prey the predator's death rate decays exponentially; and, the prey's contribution to the predator's growth rate is proportional to both the available prey and the size of the predator population. In figure 1, we provide a simplicial representation of the Lotka–Volterra model. The simplicial representation accounts for the underlying concepts of the model rather than the particular mathematical form of the model, as we want to avoid the specific details associated with any particular mathematical representation to allow for direct comparison with other models.

To arrive at the simplicial representation in figure 1, we identify the underlying concepts of equation (2.1): the agents are 'prey' and 'predator', corresponding to the population variables $x$ and $y$, respectively; the terms $ax$ and $-bxy$ on the right side of the prey equation correspond to 'prey growth' and 'predation', respectively; the terms $cxy$ and $-dy$ on the right side of the predator equation correspond to 'predator growth' and 'predator death', respectively; finally, the solutions of equation (2.1) exhibit oscillatory dynamics for both populations, which we identify as 'oscillatory population' in each case. The vertices of the simplicial representation are labelled with these concepts. To form the higher-dimensional simplices, we first identify the one-dimensional interconnections of these model concepts, based on equation (2.1), as indicated in table 1. For example, 'prey' is directly connected in a conceptual

**Table 1.** Conceptual interconnections for equation (2.1). A 'Y' indicates that there is an interconnection between the two concepts.

| | prey | prey growth | predation | oscillatory population | predator | predator growth | predator death |
|---|---|---|---|---|---|---|---|
| prey | | Y | Y | Y | | | |
| prey growth | Y | | Y | Y | | | |
| predation | Y | Y | | Y | | | |
| oscillatory population | Y | Y | Y | | Y | Y | Y |
| predator | | | Y | Y | | Y | Y |
| predator growth | | | Y | Y | Y | | Y |
| predator death | | | | Y | Y | Y | |

manner with 'prey growth', 'predation' and 'oscillatory population', where each pair of these concepts forms a 1-simplex, each subset of three of these four concepts forms a 2-simplex, and all four of these concepts form a 3-simplex. Note that 'predation' and 'predator death' have no direct connection in the simplicial representation since the effect of predator death on predation is indirect through predator growth.

Consideration of the conceptual basis of models allows for constructive comparison between models irrespective of their mathematical form. Since we generally need to compare models with different forms, we construct simplicial representations that contain the essential components of the models and their interconnections—components here naturally depend on the scientific problem studied. For instance, we may be interested in comparing a continuum model and a discrete model of the same system in terms of the components of the system that are identified in each model, in order to understand how the models are related. The simplicial representation in figure 1 is such a conceptual representation of the Lotka–Volterra model, whereby the simplicial representation provides the essential concepts and interactions underlying the model: the vertices correspond to the underlying concepts of the model, the edges link concepts that are directly interconnected, and the 2- and 3-simplices indicate higher-dimensional interconnections between related concepts.

## 2.3. Homology of simplicial representations

Given a simplicial representation of a model, we employ persistent homology to provide a unique and visual characterization of the model. Note that simplicial homology of the full simplicial complex, while simpler than persistent homology, is not applicable here since it does not in general provide a unique characterization of a model. Indeed, simplicial homology calculates the number of $k$-dimensional holes of the full simplicial complex, for nonnegative integers $k$, and the numbers of $k$-dimensional holes can be equal for distinctly labelled complexes. A simple example is given by two simplicial complexes that are each just a 0-simplex, where the 0-simplex has a different label in each complex. Then the two labelled complexes are distinct, however, they have the same numbers of $k$-dimensional holes since they both consist of a single component, or 0-dimensional hole, and have no higher-dimensional holes. We therefore require the filtration in persistent homology in order to uniquely characterize distinct simplicial complexes. This is different from the situation typically encountered in topological data analysis.

To compare a collection of simplicial representations of models using persistent homology, we require filtrations for each complex that are compatible, so that the same topological structures in different complexes give the same persistence intervals. For this, we use a reference complex such that the simplicial complexes associated with all of the models for comparison are subcomplexes.

**Notation 2.4 (Reference complex).** Let $\mathcal{C}$ be the set of all components that appear in the collection of models under consideration for comparison, and let $m$ be the maximum dimension of the simplicial complexes associated with the models. Denote by $\mathcal{R}$ the $(|\mathcal{C}| - 1)$-simplex spanned by the labelled

0-simplices $\{v_1, v_2, \ldots, v_{|\mathcal{C}|}\}$. The *reference complex* is the $m$-skeleton of $\mathcal{R}$, therefore $\mathcal{R}^{(m)}$. Note that each simplicial representation of a model under consideration is a subcomplex of the reference complex.

Therefore, to obtain filtrations for the simplicial complexes that allow for direct comparison of the persistence barcodes, we first obtain a filtration for the reference complex $\mathcal{R}^{(m)}$. The filtration of each simplicial representation is then an induced filtration of the filtration for $\mathcal{R}^{(m)}$. In this way, we can directly compare the persistence intervals between different models, since identical persistence intervals will arise from identical topological features associated with the same subset of model components.

Note that we could always use the simplex $\mathcal{R}$ as the reference complex, however, it is computationally efficient to use the $m$-skeleton $\mathcal{R}^{(m)}$ where possible. If we then need to compare an additional model generated by a subset of $\mathcal{C}$ whose simplicial representation has dimension higher than $m$, we can simply extend the filtration of the reference simplex to a filtration of a higher-dimensional skeleton of $\mathcal{R}$ (see proposition 2.13).

We want the persistent homology of a particular simplicial complex to be unique among all of the simplicial complexes of the models of interest, in order to reflect all differences between the complexes and hence models. We therefore employ a *flat filtration* ([32], ch. 11, p. 83) whereby only one simplex is added at each filtration index, and each simplex is added after all of its proper faces.

**Definition 2.5 (Flat filtration).** Let $\mathcal{C}$ be the set of all components that appear in the collection of models under consideration for comparison, and let $\mathcal{R}^{(m)}$ be the associated reference complex. Furthermore, let $w : \mathcal{R}^{(m)} \to [|\mathcal{R}^{(m)}|]$ be a bijective weight function such that if $\tau$, $\sigma \in \mathcal{R}^{(m)}$ and $\tau$ is a proper face of $\sigma$ then $w(\tau) < w(\sigma)$. The *flat filtration* of $(\mathcal{R}^{(m)}, w)$ is the nested sequence of subcomplexes $\{R_i\}_{i=1}^{|\mathcal{R}^{(m)}|}$, where $R_i = w^{-1}((-\infty, i])$ for $i \in [|\mathcal{R}^{(m)}|]$.

The flat filtration on $\mathcal{R}^{(m)}$ induces a filtration on each subcomplex of $\mathcal{R}^{(m)}$:

**Definition 2.6 (Induced filtration of a subcomplex of the reference complex).** Let $\mathcal{C}$ be the set of all components that appear in the collection of models under consideration for comparison, and let $(\mathcal{R}^{(m)}, w)$ be the associated reference complex with a flat filtration $\{R_i\}_{i=1}^{|\mathcal{R}^{(m)}|}$. Furthermore, let $L$ be a simplicial representation of a model generated by a subset of components from $\mathcal{C}$. Then the *induced filtration* for the subcomplex $L \subseteq \mathcal{R}^{(m)}$ is the nested sequence of subcomplexes $\{F_i\}_{i \in I}$ where, for each $i \in I$, $F_i = w|_L^{-1}((-\infty, i]) = R_i \cap L \neq \emptyset$, and $i \leq j$ for all $j$ such that $1 \leq j \leq |\mathcal{R}^{(m)}|$ and $R_i \cap L = R_j \cap L$.

Note that for the induced filtration we exclude filtration indices $i$ for which $R_i \cap L = \emptyset$, and since we may have $R_i \cap L = R_j \cap L$ for some filtration indices $i \neq j$ we take the smallest such index as the corresponding filtration index. Note further that the induced filtration adds one simplex at each index of the filtration, which allows us to use it as a basis for a distance between models (given the set of model components $\mathcal{C}$) as we show below (see definition 2.16 and theorem 2.18).

We now establish that the simplicial representations and associated filtrations provide the desired descriptions of the models. The following theorem shows that the simplicial complex associated with a given model has a unique multiset of persistence intervals with respect to the induced filtration. Therefore, the persistence intervals, visualized as a persistence barcode, provide a unique 'fingerprint' of the model.

**Theorem 2.7.** *Let $\mathcal{C}$ be the set of all components that appear in the collection of models under consideration for comparison, and let $\mathcal{R}^{(m)}$ be the associated reference complex. Let $K$ and $L$ be two labelled simplicial complexes associated with two models generated by subsets of $\mathcal{C}$, and let $K$ and $L$ have the induced filtrations from a flat filtration on $\mathcal{R}^{(m)}$. Then $K = L$ if and only if $\mathcal{P}_k(K) = \mathcal{P}_k(L)$ for all $k \geq 0$.*

*Proof.* The forward implication is trivial so consider the backward implication, for which we provide a contrapositive proof. Suppose that $K \neq L$, and let $\sigma^p$ be a simplex of smallest dimension $p$ in the symmetric difference $K \triangle L$. Without loss of generality, we may assume that $\sigma^p \in K$. We show that there exists $k \geq 0$ such that $\mathcal{P}_k(K) \neq \mathcal{P}_k(L)$.

If $p = 0$ then $\mathcal{P}_0(K)$ contains an interval with left endpoint equal to the filtration index $i$ at which $\sigma^p$ is added to the filtered complex, corresponding to the creation of a 0-dimensional homology class. Since $\sigma^p \notin L$ there is no simplex added at index $i$ in the filtration of $L$, hence no interval with left endpoint $i$ in $\mathcal{P}_0(L)$.

For $p \geq 1$, the boundary of $\sigma^p$ is a $(p-1)$-cycle $\tau^{p-1}$, which is a non-bounding cycle in the filtered complex prior to the addition of $\sigma^p$. So either $\tau^{p-1}$ corresponds to a linearly independent $(p-1)$-dimensional homology class, or the homology class for $\tau^{p-1}$ is a linear combination of the $(p-1)$-

dimensional homology classes. If $\tau^{p-1}$ is in a linearly independent homology class then $\mathcal{P}_{p-1}(K)$ contains an interval with right endpoint equal to the index $j$ of the filtration where the homology class is annihilated by the addition of $\sigma^p$. Since $\sigma^p \notin L$ there is no interval in $\mathcal{P}_{p-1}(L)$ with the same right endpoint $j$. Alternatively, suppose the homology class for $\tau^{p-1}$ is a linear combination of the $(p-1)$-dimensional homology classes. If the addition of $\sigma^p$ at some index $i$ in the filtration creates a $p$-cycle, and hence a new $p$-dimensional homology class, containing $\sigma^p$ then there is a corresponding interval in $\mathcal{P}_p(K)$ with left endpoint $i$ and no interval in $\mathcal{P}_p(L)$ with left endpoint $i$. Otherwise, if the addition of $\sigma^p$ at some index $i$ in the filtration does not create a $p$-cycle then, since $\tau^{p-1}$ is now a bounding cycle, it follows that one of the previously linearly independent $(p-1)$-dimensional homology classes is now linearly dependent and therefore annihilated at index $i$. Therefore, there is a corresponding interval in $\mathcal{P}_{p-1}(K)$ with right endpoint $i$, and no interval in $\mathcal{P}_{p-1}(L)$ with right endpoint $i$. ∎

We can rephrase theorem 2.7 in terms of the full multisets of persistence intervals:

**Corollary 2.8.** *Let $\mathcal{C}$ be the set of all components that appear in the collection of models under consideration for comparison, and let $\mathcal{R}^{(m)}$ be the associated reference complex. Let $K$ and $L$ be two labelled simplicial complexes associated with two models generated by subsets of $\mathcal{C}$, and let $K$ and $L$ have the induced filtrations from a flat filtration on the reference simplex $\mathcal{R}^{(m)}$. Then $K = L$ if and only if $\mathcal{P}(K) = \mathcal{P}(L)$.*

*Proof.* The forward implication follows from theorem 2.7. The backward implication will also follow from theorem 2.7 if we establish that $\mathcal{P}(K) = \mathcal{P}(L)$ if and only if $\mathcal{P}_k(K) = \mathcal{P}_k(L)$ for all $k \geq 0$. Note that $\mathcal{F}(K) = \{\mathcal{P}_k(K) \mid \mathcal{P}_k(K) \neq \emptyset$ and $k \geq 0\}$ is a partition of $\mathcal{P}(K)$, and similarly for $\mathcal{F}(L)$. So $(a, b) \in \mathcal{P}(K)$ implies $(a, b) \in \mathcal{P}_k(K)$ for some $k \geq 0$, and if $(a, b) \in \mathcal{P}(L)$ then $(a, b) \in \mathcal{P}_j(L)$ for some $j \geq 0$. Since both $K$ and $L$ are filtered with induced filtrations, whereby at most one simplex is added at each filtration index, the dimension of the simplex added at index $a$ uniquely determines the dimension of the homology class created at index $a$ corresponding to the interval $(a, b)$. Therefore, $k = j$, and it follows that $\mathcal{P}(K) = \mathcal{P}(L)$ is equivalent to $\mathcal{P}_k(K) = \mathcal{P}_k(L)$ for all $k \geq 0$. ∎

For a general filtration, the multiplicity of each persistence interval may be greater than one. We now show that with the induced filtration, however, the multiplicity of each persistence interval is equal to one, so that the multiset of persistence intervals is a set.

**Proposition 2.9.** *Let $\mathcal{C}$ be the set of all components that appear in the collection of models under consideration for comparison, and let $\mathcal{R}^{(m)}$ be the associated reference complex. Let $K$ be a labelled simplicial complex associated with a model generated by a subset of $\mathcal{C}$, and let $K$ have the induced filtration from a flat filtration on the reference complex $\mathcal{R}^{(m)}$. Then every interval in $\mathcal{P}(K)$ has multiplicity one.*

*Proof.* It suffices to show that no two intervals in $\mathcal{P}(K)$ have the same left endpoint. So suppose to the contrary that the two intervals $I_1$ and $I_2$ in the multiset $\mathcal{P}(K)$ have the same left endpoint.

Note that $I_1$ and $I_2$ do not correspond to homology classes in different dimensions. Indeed, one $p$-cycle and one $q$-cycle with $p < q$ cannot be created at the same index as this would require the simultaneous addition of a $(p-1)$-simplex and a $(q-1)$-simplex, respectively, which is not possible with the induced filtration.

So $I_1$ and $I_2$ must correspond to linearly independent homology classes in the same dimension, and therefore to the creation of two linearly independent non-bounding $p$-cycles $\sigma_1$ and $\sigma_2$ at the same filtration index $i$ due to the addition of a $p$-simplex $\tau$. Then $\sigma_1 + \sigma_2$ under $\mathbb{Z}/2\mathbb{Z}$ addition is a non-bounding cycle at index $i-1$, so the addition of $\tau$ at index $i$ yields one new homology class, contradicting the creation of two homology classes. Therefore, there is at most one interval in $\mathcal{P}(K)$ with a particular left endpoint. ∎

While we have established that the multiset of persistence intervals is a set, we continue to describe it as a multiset for clarity of reference.

Since a submodel consists of a subset of the components and interconnections of a given model, we expect a similar relationship to hold between the simplicial representations and the corresponding persistent homology, which we now confirm.

**Proposition 2.10.** *Let $\mathcal{C}$ be the set of all components that appear in the collection of models under consideration for comparison, and let $\mathcal{R}^{(m)}$ be the associated reference complex. Let $M$ and $N$ be two models for comparison, where $M$ is a submodel of $N$, and let $K$ and $L$ be the associated labelled simplicial complexes of $M$ and $N$, respectively, generated by subsets of $\mathcal{C}$. Let $K$ and $L$ have the induced filtrations from a flat filtration on $\mathcal{R}^{(m)}$. Then $K$ is a subcomplex of $L$. Furthermore, for each $k \geq 0$, the submultiset of finite intervals from $\mathcal{P}_k(K)$*

is a submultiset of $\mathcal{P}_k(L)$, and each infinite interval in $\mathcal{P}_k(K)$ is either in $\mathcal{P}_k(L)$ or corresponds to a finite interval in $\mathcal{P}_k(L)$ with the same left endpoint.

*Proof.* Since $M$ is a submodel of $N$, all of the model components and interconnections in $M$ are also in $N$, so the corresponding simplicial complexes satisfy $K \subseteq L$.

Now, fix $k \geq 0$ and let $(a, b)$ be a finite interval in $\mathcal{P}_k(K)$ corresponding to the creation and subsequent annihilation of a $k$-homology class at indices $a$ and $b$, respectively, in the filtration. Since $K \subseteq L$, the subcomplex of $K$ corresponding to the $k$-homology class is also a subcomplex of $L$, so $(a, b) \in \mathcal{P}_k(L)$.

Suppose now that the infinite interval $(a, \infty)$ is in $\mathcal{P}_k(K)$, corresponding to the creation of a $k$-homology class at index $a$ in the filtration which is not annihilated and persists in the full complex $K$. Suppose further, without loss of generality, that $(a, \infty) \notin \mathcal{P}_k(L)$. Since $K \subseteq L$, the subcomplex of $K$ corresponding to the $k$-homology class is also a subcomplex of $L$, so the $k$-homology class is also created in the filtration of $L$. Since $(a, \infty) \notin \mathcal{P}_k(L)$, $L$ contains the $(k+1)$-simplex for which the $k$-homology class is the bounding cycle, so the $k$-homology class is annihilated at some index $b$ in the filtration of $L$ due to the addition of the $(k+1)$-simplex. Thus, $(a, b) \in \mathcal{P}_k(L)$. ∎

We visualize the persistent homology of a simplicial complex, summarized by the multisets of persistence intervals, as a persistence barcode. The barcodes have the distinct advantage of allowing for a visual comparison of models.

**Definition 2.11 (Persistence barcode).** Let $\mathcal{C}$ be the set of all components that appear in the collection of models under consideration for comparison, and let $\mathcal{R}^{(m)}$ be the associated reference complex. Let $K$ be a labelled simplicial complex associated with a model generated by a subset of $\mathcal{C}$, and let $K$ have the induced filtration from a flat filtration on the reference complex $\mathcal{R}^{(m)}$. For each dimension $p \geq 0$, the $p$th *persistence barcode* $\mathrm{BAR}_p(K)$ for $K$ is a graphical representation of the multiset of $p$-dimensional persistence intervals $\mathcal{P}_p(K)$. Specifically, let $f : \mathcal{P}_p(K) \to [|\mathcal{P}_p(K)|]$ be a bijection, then $\mathrm{BAR}_p(K)$ is given by the set of points $\{(x, f(a, b)) \mid (a, b) \in \mathcal{P}_p(K) \text{ and } x \in [a, b)\}$, which is unique up to the bijection specifying the order with respect to the second coordinate.

While two different flat filtrations of the reference complex $\mathcal{R}^{(m)}$ can produce two different multisets of persistence intervals for a subcomplex $K$, the numbers of finite and infinite intervals in each submultiset $\mathcal{P}_k(K)$ are unchanged.

**Proposition 2.12.** *Let $\mathcal{C}$ be the set of all components that appear in the collection of models under consideration for comparison, and let $\mathcal{R}^{(m)}$ be the associated reference complex. Let $K$ be a labelled simplicial complex associated with a model generated by a subset of components in $\mathcal{C}$, let $\mathcal{F}$ and $\mathcal{F}'$ be two induced filtrations of $K$ corresponding to two different flat filtrations of the reference complex $\mathcal{R}^{(m)}$, and let $\mathcal{P}(K)$ and $\mathcal{P}'(K)$ be the corresponding multisets of persistence intervals. Then, for all $k \geq 0$, $|\mathcal{P}_k(K)| = |\mathcal{P}'_k(K)|$, and in particular the numbers of finite and infinite intervals are the same in $\mathcal{P}_k(K)$ and $\mathcal{P}'_k(K)$.*

*Proof.* Fix $k \geq 0$. We first establish a bijection $f : \mathcal{P}_k(K) \to \mathcal{P}'_k(K)$. Let $I \in \mathcal{P}_k(K)$, with left endpoint $i$ corresponding to the creation of a non-bounding $k$-cycle $\sigma^k$ at index $i$ of filtration $\mathcal{F}$. Since the $k$-cycle is also created at some index $j$ of filtration $\mathcal{F}'$, there exists $I' \in \mathcal{P}'_k(K)$ with left endpoint $j$, so we define $f(I) = I'$. The mapping $f$ is well defined, since each topological feature appears at a unique filtration index.

We need to show that $f$ is injective and surjective. For injectivity, suppose that $I_1, I_2 \in \mathcal{P}_k(K)$ with $f(I_1) = f(I_2)$. In particular, the intervals $f(I_1)$ and $f(I_2)$ have the same left endpoint corresponding to the creation of a non-bounding $k$-cycle in the filtration $\mathcal{F}'$, and since this non-bounding $k$-cycle is created at the left endpoints of both $I_1$ and $I_2$ we have $I_1 = I_2$. For surjectivity, let $I' \in \mathcal{P}'_k(K)$. Then the left endpoint of $I'$ corresponds to the creation of a non-bounding $k$-cycle in the filtration $\mathcal{F}'$, and this non-bounding $k$-cycle is also created at some index $i$ of the filtration $\mathcal{F}'$, so there exists $I \in \mathcal{P}_k(K)$ with left endpoint $i$, so that $f(I) = I'$.

Finally, the two induced filtrations correspond to the same simplicial complex $K$, so $\mathcal{P}_k(K)$ and $\mathcal{P}'_k(K)$ have the same number of infinite intervals, and therefore the same number of finite intervals. ∎

We can extend the model comparison to include new models for which not all components are in $\mathcal{C}$ by appending the new components to $\mathcal{C}$ to obtain a superset $\overline{\mathcal{C}}$, and then extending the ordering function on $\mathcal{C}$ to $\overline{\mathcal{C}}$. This may be necessary as our set of candidate models grows, or if we want to increase the level of detail in which we describe models. This extension method preserves the simplicial representations, and associated persistence intervals, of the models generated by $\mathcal{C}$. Notationally, we use an overline to denote concepts in the extended system.

**Proposition 2.13.** *Let $C$ and $\overline{C}$ be two sets of model components with $\overline{C} \supseteq C$. Let $\overline{\mathrm{Ord}} : \overline{C} \to [|\overline{C}|]$ be a bijection specifying an order for $\overline{C}$ such that $\overline{\mathrm{Ord}}|_C = \mathrm{Ord}$. Further, let $\mathcal{R}^{(m)}$ and $\overline{\mathcal{R}}^{(m)}$ be the reference complexes associated with $C$ and $\overline{C}$, respectively. Then we have the following:*

- (i) *The reference complexes satisfy $\mathcal{R}^{(m)} \subseteq \overline{\mathcal{R}}^{(m)}$.*
- (ii) *Each flat filtration of $\mathcal{R}^{(m)}$ can be extended to a flat filtration of $\overline{\mathcal{R}}^{(m)}$.*
- (iii) *Suppose $\mathcal{R}^{(m)}$ has a flat filtration, which is extended to a flat filtration on $\overline{\mathcal{R}}^{(m)}$. For a simplicial representation $K$ of a model generated by a subset of components in $C$ we have $\mathcal{P}_k(K) = \overline{\mathcal{P}}_k(K)$ for all $k \geq 0$. In other words, the persistent homology for $K$ corresponding to the induced filtration from the flat filtration on $\mathcal{R}^{(m)}$ is the same as the persistent homology for $K$ corresponding to the induced filtration from the extended flat filtration on $\overline{\mathcal{R}}^{(m)}$.*

*Proof.* For (i), note that $\mathcal{R}^{(m)}$ is spanned by a subset of the 0-simplices that span $\overline{\mathcal{R}}^{(m)}$, so $\mathcal{R}^{(m)}$ is a subcomplex of $\overline{\mathcal{R}}^{(m)}$.

For (ii), let $w : \mathcal{R}^{(m)} \to [|\mathcal{R}^{(m)}|]$ be a bijective weight function such that if $\tau$ is a proper face of $\sigma$ then $w(\tau) < w(\sigma)$, yielding a flat filtration $\{R_i\}_{i=1}^{|\mathcal{R}^{(m)}|}$ where each $R_i = w^{-1}((-\infty, i])$. Extend $w$ to a bijective weight function $\overline{w} : \overline{\mathcal{R}}^{(m)} \to [|\overline{\mathcal{R}}^{(m)}|]$ such that if $\tau$ is a proper face of $\sigma$ then $\overline{w}(\tau) < \overline{w}(\sigma)$. Then the flat filtration $\{\overline{R}_i\}_{i=1}^{|\overline{\mathcal{R}}^{(m)}|}$ of $\overline{\mathcal{R}}^{(m)}$, where $\overline{R}_i = \overline{w}^{-1}((-\infty, i])$, extends the flat filtration of $\mathcal{R}^{(m)}$, that is, $R_i = \overline{R}_i$ for $i \in [|\mathcal{R}^{(m)}|]$.

For (iii), let $K$ be a simplicial representation of a model generated by a subset of components in $C$. Since the flat filtration of $\mathcal{R}^{(m)}$ and the extended flat filtration of $\overline{\mathcal{R}}^{(m)}$ are equal over the first $|\mathcal{R}^{(m)}|$ filtration indices, the corresponding induced filtrations are equal, and it follows that $\mathcal{P}_k(K) = \overline{\mathcal{P}}_k(K)$ for all $k \geq 0$. ∎

## 2.4. Model comparison by distance

To compare models of interest, given a set of model components $C$, in a quantitative manner we introduce a measure of distance between the simplicial representations. We provide a direct measure between the simplicial complexes, along with a measure based on the corresponding multisets of persistence intervals, and then show that the two measures yield the same distance.

**Definition 2.14 (Distance between simplicial complexes).** Let $C$ be the set of all components that appear in the collection of models under consideration for comparison. Let $\mathcal{S}$ be a collection of labelled simplicial complexes corresponding to models generated by subsets from $C$. Define the function

$$d_C : \mathcal{S} \times \mathcal{S} \to \mathbb{R} \quad \text{such that } d_C(K, L) = |K \triangle L|, \tag{2.2}$$

so that $d_C$ gives the cardinality of the symmetric difference of the two labelled simplicial complexes.

Some important remarks regarding the distance function between simplicial complexes:

— The symmetric difference of two labelled simplicial complexes accounts for the labelling of the simplices, so that two labelled simplices from different simplicial complexes are considered identical when they are spanned by the same set of labelled 0-simplices.

— The function $d_C$ calculates the number of labelled simplices that must be added to or subtracted from one of the simplicial complexes to obtain the other simplicial complex. In particular, this measure of distance applies for any two simplicial representations, whether or not they have labelled simplices in common.

— The function $d_C$ is invariant with respect to a reordering of the model components in $C$, and therefore a relabelling of the vertices of the reference complex, since the original and relabelled simplicial complexes are isomorphic.

— We label the distance function $d_C$ with the subscript $C$ to emphasize that the distance between two simplicial representations is dependent on the level of conceptual detail employed according to the chosen set of model components $C$.

— Since we add simplices under the field $\mathbb{Z}/2\mathbb{Z}$ we have that $K \triangle L = K + L$.

**Notation 2.15.** For a multiset $S$ of intervals $(a, b) \in \mathbb{N}^* \times \overline{\mathbb{N}}^*$, we denote the projection onto the first coordinates as $\mathrm{proj}_1(S) = \{a \in \mathbb{N}^* \mid (a, b) \in S \text{ for some } b \in \overline{\mathbb{N}}^*\}$, and the projection onto the second coordinates as $\mathrm{proj}_2(S) = \{b \in \overline{\mathbb{N}}^* \mid (a, b) \in S \text{ for some } a \in \mathbb{N}^*\}$.

**Definition 2.16 (Distance between multisets of persistence intervals).** Let $\mathcal{C}$ be the set of all components that appear in the collection of models under consideration for comparison, and let $\mathcal{R}^{(m)}$ be the associated reference complex with a flat filtration. Let $\mathcal{M}$ be a collection of multisets of persistence intervals $\mathcal{P}(K)$ corresponding to labelled simplicial complexes $K$ with labels from $\mathcal{C}$ and induced filtrations from the flat filtration. Furthermore, let

$$\Theta : \mathcal{M} \to 2^{[|\mathcal{R}^{(m)}|]} \quad \text{be such that } \Theta(\mathcal{P}(K)) = \text{proj}_1(\mathcal{P}(K)) \cup (\text{proj}_2(\mathcal{P}(K)) \setminus \{+\infty\}), \tag{2.3}$$

which is the set of all left endpoints and finite right-endpoints of the persistence intervals. Define the function

$$\hat{d}_{\mathcal{C}} : \mathcal{M} \times \mathcal{M} \to \mathbb{R} \quad \text{by } \hat{d}_{\mathcal{C}}(\mathcal{P}(K), \mathcal{P}(L)) = |\Theta(\mathcal{P}(K)) \triangle \Theta(\mathcal{P}(L))|. \tag{2.4}$$

We now show that $d_{\mathcal{C}}$ is a distance function on $\mathcal{S}$, that $\hat{d}_{\mathcal{C}}$ is a distance function on $\mathcal{M}$, and that the metric spaces $(\mathcal{S}, d)$ and $(\mathcal{M}, \hat{d}_{\mathcal{C}})$ are isometric, so we can use either metric to determine the distances between simplicial complexes. To prove these claims, we use the triangle inequality for the symmetric difference: for sets $X$, $Y$ and $Z$, the cardinality of the symmetric difference satisfies subadditivity, that is, $|X \triangle Z| \leq |X \triangle Y| + |Y \triangle Z|$. This relation follows by observing that $X \triangle Z \subseteq (X \triangle Y) \cup (Y \triangle Z)$.

We first need a lemma to show that the function $\hat{d}_{\mathcal{C}}$ is invariant with respect to a change in the flat filtration on the reference complex.

**Lemma 2.17.** *Let $\mathcal{C}$ be the set of all components that appear in the collection of models under consideration for comparison, and let $\mathcal{R}^{(m)}$ be the associated reference complex. The function $\hat{d}_{\mathcal{C}}$ is invariant with respect to a change in the flat filtration of the reference complex $\mathcal{R}^{(m)}$.*

*Proof.* Let $\mathcal{P}(K)$ and $\mathcal{P}(L)$ be the multisets of persistence intervals for $K$ and $L$, respectively, corresponding to the induced filtrations from the first flat filtration with associated weight function $w$, and let $\mathcal{P}'(K)$ and $\mathcal{P}'(L)$ be the multisets of persistence intervals for $K$ and $L$, respectively, corresponding to the induced filtrations from the second flat filtration with associated weight function $w'$.

There exists a permutation $\pi$ on $[|\mathcal{R}^{(m)}|]$ such that $\pi \circ w = w'$, from which it follows that there is a bijection $f : \Theta(\mathcal{P}(K)) \cup \Theta(\mathcal{P}(L)) \to \Theta(\mathcal{P}'(K)) \cup \Theta(\mathcal{P}'(L))$ where the restriction of $f$ to $\Theta(\mathcal{P}(K))$ is a bijection onto $\Theta(\mathcal{P}'(K))$ and the restriction of $f$ to $\Theta(\mathcal{P}(L))$ is a bijection onto $\Theta(\mathcal{P}'(L))$. It follows that there is a bijection between $\Theta(\mathcal{P}(K)) \triangle \Theta(\mathcal{P}(L))$ and $\Theta(\mathcal{P}'(K)) \triangle \Theta(\mathcal{P}'(L))$, so $\hat{d}_{\mathcal{C}}(\mathcal{P}(K), \mathcal{P}(L)) = \hat{d}_{\mathcal{C}}(\mathcal{P}'(K), \mathcal{P}'(L))$. ∎

**Theorem 2.18.** *Let $\mathcal{C}$ be the set of all components of models under consideration, and let $\mathcal{R}^{(m)}$ be the associated reference complex with a flat filtration. Let $\mathcal{S}$ be a collection of labelled simplicial complexes with labels from $\mathcal{C}$, and let $\mathcal{M}$ be the collection of multisets of persistence intervals $\mathcal{P}(K)$, for $K \in \mathcal{S}$ with the induced filtrations. Then the function $d_{\mathcal{C}}$ is a metric on $\mathcal{S}$, the function $\hat{d}_{\mathcal{C}}$ is a metric on $\mathcal{M}$, and the metric spaces $(\mathcal{S}, d)$ and $(\mathcal{M}, \hat{d}_{\mathcal{C}})$ are isometric.*

*Proof.* To see that $d_{\mathcal{C}}$ is a metric on $\mathcal{S}$, suppose that $K$, $L \in \mathcal{S}$. Then $d_{\mathcal{C}}(K, L) = 0$ if and only if $|K \triangle L| = 0$ if and only if $K = L$, so the identity of indiscernibles holds. Symmetry follows from the symmetric difference. It remains to show subadditivity, so let $M \in \mathcal{S}$. The triangle inequality for the symmetric difference gives $d_{\mathcal{C}}(K, L) = |K \triangle L| \leq |K \triangle M| + |M \triangle L| = d_{\mathcal{C}}(K, M) + d_{\mathcal{C}}(M, L)$, as required.

Next, we show that $\hat{d}_{\mathcal{C}}$ is a metric on $\mathcal{M}$, noting that we can consider any particular induced filtration since $\hat{d}_{\mathcal{C}}$ is invariant with respect to a change in the flat filtration of the reference complex by lemma 2.17. So let $\mathcal{P}(K)$, $\mathcal{P}(L) \in \mathcal{M}$ be two multisets of persistence intervals corresponding to the simplicial complexes $K$ and $L$. For the identity of indiscernibles, $\hat{d}_{\mathcal{C}}(\mathcal{P}(K), \mathcal{P}(L)) = 0$ if and only if $\Theta(\mathcal{P}(K)) = \Theta(\mathcal{P}(L))$, so that $K$ and $L$ contain the same simplices, if and only if $K = L$ if and only if $\mathcal{P}(K) = \mathcal{P}(L)$ by corollary 2.8. Symmetry follows from the symmetric difference, and subadditivity follows from the triangle inequality for the symmetric difference.

Finally, to show that the two metric spaces $(\mathcal{S}, d_{\mathcal{C}})$ and $(\mathcal{M}, \hat{d}_{\mathcal{C}})$ are isometric we need to establish a surjective isometry $f : \mathcal{S} \to \mathcal{M}$. Define $f$ such that $K \mapsto \mathcal{P}(K)$, which is a surjection. Recall that $f$ is an isometry if and only if $\hat{d}_{\mathcal{C}}(f(K), f(L)) = d_{\mathcal{C}}(K, L)$ for all $K$, $L \in \mathcal{S}$, or equivalently the cardinality of $K \triangle L$ equals the cardinality of $\Theta(\mathcal{P}(K)) \triangle \Theta(\mathcal{P}(L))$ for all $K$, $L \in \mathcal{S}$. To show that $f$ is an isometry it therefore suffices to establish a bijection $\phi_{(K,L)}$, for each $K$, $L \in \mathcal{S}$, from $K \triangle L$ onto $\Theta(\mathcal{P}(K)) \triangle \Theta(\mathcal{P}(L))$. Let $g : \mathcal{R}^{(m)} \to [|\mathcal{R}^{(m)}|]$ be the bijective weight function for the flat filtration of the reference complex.

Define the function $\phi_{(K,L)} = g|_{K \triangle L}$ by restriction. Then $\phi_{(K,L)}$ is a bijection onto its image, since simplices are added individually in the flat filtration. It remains to show that the image of $K \triangle L$ under $\phi_{(K,L)}$ is $\Theta(\mathcal{P}(K)) \triangle \Theta(\mathcal{P}(L))$. Indeed, $\sigma \in K \triangle L$ if and only if $a \in \Theta(\mathcal{P}(K)) \triangle \Theta(\mathcal{P}(L))$ where $g|_{K \triangle L}(\sigma) = a$. ∎

If we know the relationships between two pairs of simplicial representations with a complex in common, say between $J$ and $K$ and between $K$ and $L$, then we can infer the relationship between $J$ and $L$. Specifically, the distance between $J$ and $L$ can be determined either from $J \triangle K$ and $K \triangle L$ or, if the complexes are all subcomplexes of $\mathcal{R}^{(m)}$ with a flat filtration, from the multisets of persistence intervals. Formally,

**Proposition 2.19.** *Let $\mathcal{C}$ be the set of all components of models under consideration, and let $\mathcal{R}^{(m)}$ be the associated reference complex with a flat filtration. Let $\mathcal{S}$ be a collection of labelled simplicial complexes corresponding to models generated by subsets from $\mathcal{C}$. For $J, K, L \in \mathcal{S}$ we have*

$$d_{\mathcal{C}}(J, L) = |(J \triangle K) \triangle (K \triangle L)|$$

*and*

$$\hat{d}_{\mathcal{C}}(\mathcal{P}(J), \mathcal{P}(L)) = \left| \Big(\Theta(\mathcal{P}(J)) \triangle \Theta(\mathcal{P}(K))\Big) \triangle \Big(\Theta(\mathcal{P}(K)) \triangle \Theta(\mathcal{P}(L))\Big) \right|.$$

*Proof.* Both equations follow immediately from the observation that the symmetric difference satisfies $X \triangle Z = (X \triangle Y) \triangle (Y \triangle Z)$ for sets $X$, $Y$ and $Z$. ∎

## 2.5. Model comparison by equivalence

We now consider model comparison by equivalence, which accounts for particular conceptual similarities between models that we may regard as essentially identical. So rather than comparing the simplicial complexes associated to models based on the presence or absence of a particular labelled simplex, as in model comparison by distance, here we consider the equivalence of models in terms of an equivalence of the corresponding labelled simplicial complexes. As we show below, 'equivalence' as used here is restricted to five narrowly defined operations. Moreover, equivalence between two models is only possible (but not guaranteed) for models that have high similarity in terms of the associated simplicial complexes.

For this, we need to specify a *predicate*, Pr, which is a statement that contains a finite number of variables with the specified domain Dom, such that Pr becomes a (Boolean) proposition when instantiated. We can consider a predicate to be a Boolean-valued function Pr : Dom → {true, false}. Predicates are often described in terms of the number of their variables, so that for an integer $n \geq 1$ an $n$-place predicate $\mathrm{Pr}(x_1, \ldots, x_n)$ has $n$ variables $x_1, \ldots, x_n$ with domain Dom $\subseteq D_1 \times \cdots \times D_n$, where $x_i \in D_i$ for $i = 1, \ldots, n$.

**Definition 2.20 (Equivalent simplicial complexes).** Let $\mathcal{C}$ be the set of all components of models under consideration, let $\mathcal{S}$ be a collection of labelled simplicial complexes with labels from $\mathcal{C}$, and let Pr be a two-place predicate on $\mathcal{S} \times \mathcal{S}$ such that

$$R_{\mathrm{eq}} = \{(K, L) \in \mathcal{S} \times \mathcal{S} \mid \mathrm{Pr}(K, L)\} \tag{2.5}$$

is an equivalence relation on $\mathcal{S}$. Then the simplicial complexes $K$, $L \in \mathcal{S}$ are *equivalent* if and only if $(K, L) \in R_{\mathrm{eq}}$.

To determine the similarity of two models, represented as simplicial complexes, we first need to formulate a predicate on which equivalence is based. There is not a unique predicate for this purpose of comparison, and the established equivalence of models must always be considered with regard to the particular choice of predicate. Note further that equivalence is not a quantitative comparison, but rather a qualitative observation that models have similar features with respect to the specified predicate. Informally, we want to say that $K$, $L \in \mathcal{S}$ are equivalent when their labelled simplices are conceptually the same, so in particular, we need to identify the labelled subcomplexes in $K \setminus L$ with those in $L \setminus K$. For this, we employ five operations on a labelled simplicial complex $K$. Since the 1-skeleton of a labelled simplicial complex is a labelled undirected graph, we refer to the 0-simplices and 1-simplices as vertices and edges, respectively.

We first recall the definition of a simplicial map [33,34]:

**Definition 2.21 (Vertex map, simplicial map).** Let $K$ and $L$ be two simplicial complexes. A *vertex map* is a function $\psi : K^{(0)} \to L^{(0)}$, and a *simplicial map* $\psi : K \to L$ is a function such that $\psi|_{K^{(0)}}$ is a vertex map and whenever $W = \{w_i\}_{i=0}^n$ spans a simplex in $K$ then $\psi(W) = \{\psi(w_i)\}_{i=0}^n$ spans a simplex in $L$.

We also require the notion of a simplicial multivalued map, which we define as follows:

**Definition 2.22 (Simplicial multivalued map).** Let $K$ and $L$ be two simplicial complexes, and let $F : K^{(0)} \rightrightarrows L^{(0)}$ be a left-total binary relation such that $F(w)$ is a non-empty subset of $L^{(0)}$ for each $w \in K^{(0)}$. Then $F$ is a *simplicial multivalued map*, denoted $F : K \rightrightarrows L$, if whenever a set of vertices $W = \{w_i\}_{i=0}^n$ spans a simplex in $K$ then the set of vertices $F(W) = \{F(w_i)\}_{i=0}^n$ spans a simplex in $L$, and for each $i$ we possibly have $|F(w_i)| > 1$. We say that $F$ is *absolutely injective* when $F(v) \cap F(w) \neq \emptyset$ implies $v = w$, for any $v, w \in K^{(0)}$, and $F$ is *surjective* when $F(K^{(0)}) = L^{(0)}$.

We now define the five operations on simplicial complexes. Recall that two distinct vertices in a simplicial complex are *adjacent* if they belong to the same simplex, and for a vertex $u$ in a simplicial complex $K$ we denote the set of all vertices adjacent to $u$ as $V_K(u)$, noting that $u \notin V_K(u)$.

**Definition 2.23 (Operation 1: Adjacent-vertex identification).** Let $\mathcal{C}$ be the set of all components of models under consideration, and let $K$ and $L$ be two labelled simplicial complexes with labels from $\mathcal{C}$. Let $\{u, v\}$ be a pair of adjacent vertices in $K$ such that the following all hold:

— $V_K(u) \setminus \{v\} = V_K(v) \setminus \{u\}$.
— For any non-empty subset $W \subseteq V_K(u) \setminus \{v\}$, the vertices $W \cup \{u\}$ span a simplex in $K$ if and only if the vertices $W \cup \{v\}$ span a simplex in $K$.

A simplicial map $\pi_1 : K \to L$ is an *adjacent-vertex identification* if $\pi_1$ is surjective, and is injective and label preserving on every vertex except at the pair of vertices $\{u, v\}$ that are mapped to a single vertex $c \in L^{(0)}$. That is, for $z \in K^{(0)}$,

$$\pi_1(z) = \begin{cases} z & \text{if } z \notin \{u, v\}, \\ c & \text{if } z \in \{u, v\}. \end{cases} \tag{2.6}$$

**Definition 2.24 (Operation 2: Nonadjacent-vertex identification).** Let $\mathcal{C}$ be the set of all components of models under consideration, and let $K$ and $L$ be two labelled simplicial complexes with labels from $\mathcal{C}$. Let $\{u, v\}$ be a pair of nonadjacent vertices in $K$ such that the following all hold:

— $V_K(u) = V_K(v)$.
— For any non-empty subset $W \subseteq V_K(u)$, the vertices $W \cup \{u\}$ span a simplex in $K$ if and only if the vertices $W \cup \{v\}$ span a simplex in $K$.

A simplicial map $\pi_2 : K \to L$ is a *nonadjacent-vertex identification* if $\pi_2$ is surjective, and is injective and label preserving on every vertex except at the pair of vertices $\{u, v\}$ that are mapped to a single vertex $c \in L^{(0)}$. That is, for $z \in K^{(0)}$,

$$\pi_2(z) = \begin{cases} z & \text{if } z \notin \{u, v\}, \\ c & \text{if } z \in \{u, v\}. \end{cases} \tag{2.7}$$

**Definition 2.25 (Operation 3: Vertex split).** Let $\mathcal{C}$ be the set of all components of models under consideration, and let $K$ and $L$ be two labelled simplicial complexes with labels from $\mathcal{C}$. Furthermore, let $u \in K^{(0)}$ and $c, d \in L^{(0)} \setminus K^{(0)}$ be three vertices with mutually distinct labels such that the following all hold:

— The vertices $c, d \in L$ are adjacent.
— $V_L(c) \setminus \{d\} = V_L(d) \setminus \{c\}$.
— If $W \subseteq V_K(u)$ is a nonempty subset such that the vertices $W \cup \{u\}$ span a simplex in $K$ then there exists a subset $X \subseteq V_L(c) \setminus \{d\}$ such that $|W| = |X|$, the vertices in $W$ and $X$ have the same labels, the vertices $X \cup \{c\}$ span a simplex in $L$, and the vertices $X \cup \{d\}$ span a simplex in $L$.
— If $X \subseteq V_L(c) \setminus \{d\}$ is a non-empty subset such that the vertices $X \cup \{c\}$ span a simplex in $L$ and the vertices $X \cup \{d\}$ span a simplex in $L$ then there exists a subset $W \subseteq V_K(u)$ such that $|W| = |X|$, the vertices in $W$ and $X$ have the same labels, and the vertices $W \cup \{u\}$ span a simplex in $K$.

A simplicial multivalued map $\pi_3 : K \rightrightarrows L$ is a *vertex split* if $\pi_3$ is surjective, absolutely injective, and single valued and label preserving on every vertex except at the vertex $u$ which is mapped to the pair of vertices $\{c, d\}$. That is, for $z \in K^{(0)}$,

$$\pi_3(z) = \begin{cases} z & \text{if } z \neq u, \\ \{c, d\} & \text{if } z = u. \end{cases} \tag{2.8}$$

**Definition 2.26 (Operation 4: Inclusion).** Let $\mathcal{C}$ be the set of all components of models under consideration, and let $K$ and $L$ be two labelled simplicial complexes with labels from $\mathcal{C}$. A simplicial map $\pi_4 : K \to L$ is an *inclusion* if $\pi_4$ is injective and preserves all labels. That is, for $z \in K^{(0)}$, $\pi_4(z) = z$.

**Definition 2.27 (Operation 5: Vertex substitution).** Let $\mathcal{C}$ be the set of all components of models under consideration, and let $K$ and $L$ be two labelled simplicial complexes with labels from $\mathcal{C}$. A simplicial map $\pi_5 : K \to L$ is a *vertex substitution* if $\pi_5$ is bijective and preserves all labels except for one whereby the labelled vertex $u \in K^{(0)}$ is mapped to the labelled vertex $c \in L^{(0)}$. That is, for $z \in K^{(0)}$,

$$\pi_5(z) = \begin{cases} z & \text{if } z \neq u, \\ c & \text{if } z = u. \end{cases} \tag{2.9}$$

Note that, for appropriate simplicial complexes, an adjacent-vertex identification is mutually inverse with a corresponding vertex split, a nonadjacent-vertex identification is mutually inverse with an inclusion, and a vertex substitution is mutually inverse with another vertex substitution.

To rigorously establish an equivalence between models through application of Operations 1–5, we must ensure that each operation preserves the representation of the general physical system. It is important to note that an equivalence between models is based on the components of the models that we allow to be identified as equivalent, and therefore the corresponding operations on the simplicial complexes that we regard as admissible. Thus, in addition to the formal requirements (Operations 1–5) the notion of equivalence also incorporates tight domain-specific constraints. The existence of an equivalence between models is therefore dependent on the level of model detail that we include in the simplicial representations, and the model components that we allow to be equivalent at this level of model detail. This approach provides great flexibility to compare models at various levels of conceptual detail, and whether or not models are equivalent or inequivalent will depend on the perspective with which we want to view the models. Model equivalence is not an absolute determination, but rather a perspective that is relative to the operations regarded as admissible.

The application of both Operations 1 and 3 is intrinsically restrictive, since they require similar interconnections between the vertices. Here, we explicitly specify the form in which the five operations are admissible.

**Definition 2.28 (Admissible operations on labelled simplicial complexes).** An *admissible operation* on a labelled simplicial complex associated with a model is one of the following:

(i) Operation 1 (Adjacent-vertex identification): the two identified vertices and the new vertex must have labels that represent conceptually equivalent components of the model, which should intrinsically hold for the two identified vertices since they are adjacent so have a direct interconnection.

(ii) Operation 2 (Nonadjacent-vertex identification): the two identified vertices and the new vertex must have labels that represent conceptually equivalent components of the model, which may not hold for the two identified vertices since they are not adjacent and so lack a direct interconnection.

(iii) Operation 3 (Vertex split): the original vertex and the two new vertices must have labels that represent conceptually equivalent components of the model, which should intrinsically hold for the two new vertices since they are adjacent.

(iv) Operation 4 (Inclusion): the inclusion already implies that the original simplicial complex is conceptually related to the supercomplex.

(v) Operation 5 (Vertex substitution): the new and old vertices must have labels that represent conceptually equivalent components of the model.

We are now in a position to define the required predicate.

**Definition 2.29 (Predicate for the equivalence of models).** The two-place predicate Pr on $\mathcal{S} \times \mathcal{S}$ is as follows: for $(K, L) \in \mathcal{S} \times \mathcal{S}$ there exists a sequence, which may be empty, of the five admissible operations on simplicial complexes, say $(f_i)_{i=0}^n$, such that each $f_i$ is invertible and $f_n \circ \cdots \circ f_1 \circ f_0(K) = L$.

So for $(K, L) \in \mathcal{S} \times \mathcal{S}$, the proposition $\mathrm{Pr}(K, L)$ is true when a sequence of the five admissible operations can transform $K$ to $L$ and the corresponding sequence of inverse operations can transform $L$ to $K$, otherwise if no such sequence exists then $\mathrm{Pr}(K, L)$ is false. The following theorem establishes that the specified predicate gives the necessary equivalence relation.

**Theorem 2.30.** *If* Pr *is the two-place predicate on $\mathcal{S} \times \mathcal{S}$ given in definition 2.29 then it follows that the relation $R_{\mathrm{eq}} = \{(x, y) \in \mathcal{S} \times \mathcal{S} \mid \mathrm{Pr}(x, y)\}$ is an equivalence relation on $\mathcal{S}$.*

*Proof.* $R_{\mathrm{eq}}$ is reflexive, since we can apply the empty sequence of operations to any simplicial complex. Symmetry of $R_{\mathrm{eq}}$ follows from the required invertibility of each operation, and transitivity follows from the associativity of composition of the operations. Therefore, $R_{\mathrm{eq}}$ is an equivalence relation on $\mathcal{S}$. ∎

We reiterate that the model equivalence that we consider here is quite restrictive, so only models that have a high level of similarity would be identified as equivalent. There are many other possible definitions of model equivalence that may be more appropriate when comparing different collections of models. It is important to note that our notion of equivalence of models occurs infrequently, and the demonstration of an equivalence of two models reveals fundamental and non-trivial similarities between the models.

# 3. Results and discussion

We now apply our methodology for model comparison to two different collections of models: first, we examine the model equivalence of three enzyme-catalysed reaction mechanisms involving two substrates; and second, we employ our methodology for model comparison to the two main categories of models for developmental pattern formation.

## 3.1. Comparison of bisubstrate reactions

Here, we consider the equivalence of three models for enzyme-catalysed reaction mechanisms involving two substrates: the ordered sequential bisubstrate reaction and the random sequential bisubstrate reaction, which are both ternary-complex mechanisms whereby both substrates bind to the enzyme simultaneously; and the ping-pong bisubstrate reaction, which involves a chemically modified intermediate form of the enzyme [35].

### 3.1.1. Ordered sequential bisubstrate reaction

In an ordered sequential bisubstrate reaction, two substrates $A$ and $B$ first combine with the enzyme $E$ to form a ternary complex $EAB$, followed by the reaction, and then the release of the products $P$ and $Q$. This reaction is shown schematically in figure 2*a*, along with the corresponding simplicial representation for our chosen level of conceptual detail. Note that the substrates combine with the enzyme in a particular order, here A followed by B, and the products are released in a particular order, here $P$ then Q, resulting in a single reaction path.

### 3.1.2. Random sequential bisubstrate reaction

In a random sequential bisubstrate reaction, two substrates $A$ and $B$ first combine with the enzyme $E$ to form a ternary complex $EAB$, followed by the reaction, and then the release of the products $P$ and $Q$. This reaction is shown schematically in figure 2*b*, along with the corresponding simplicial representation for our chosen level of conceptual detail. Note that, in this case, there is no required order for substrate combination with the enzyme and for product release, resulting in four possible reaction paths. This simplicial representation consists of four components, from a homological perspective, since it represents four different potential reaction paths.

### 3.1.3. Ping-pong bisubstrate reaction

In a ping-pong bisubstrate reaction, also called a double-displacement reaction, the substrate A combines with the enzyme resulting in the release of product $P$ and the formation of the intermediate $E^*$, and then

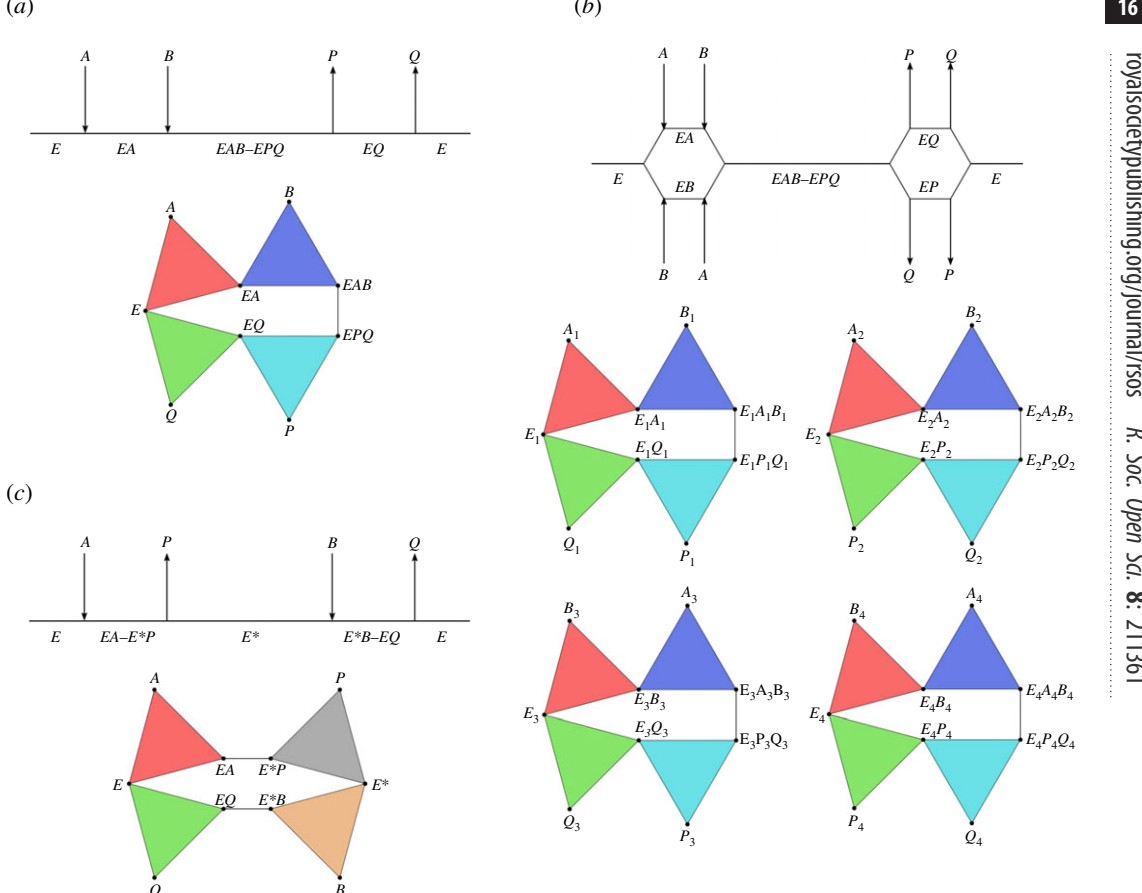

**Figure 2.** Bisubstrate reactions represented schematically and as associated labelled simplicial complexes. (*a*) Ordered sequential bisubstrate reaction, (*b*) random sequential bisubstrate reaction and (*c*) ping-pong bisubstrate reaction.

substrate $B$ combines with $E^*$ resulting in the release of product $Q$ and regeneration of the enzyme $E$. This reaction is shown schematically in figure 2$c$, along with the corresponding simplicial representation for our chosen level of conceptual detail. Note that the ping-pong mechanism requires the release of the first product before the second substrate can react.

### 3.1.4. Equivalence of bisubstrate reactions

We first demonstrate an equivalence between the ordered sequential bisubstrate reaction and the random sequential bisubstrate reaction. To transform the complex for the ordered reaction to the complex for the random reaction, apply vertex substitutions so that $E$, $A$, $EA$, $B$, $EAB$, $EPQ$, $P$, $EQ$ and $Q$ become the corresponding $E_1$, $A_1$, $E_1A_1$, $B_1$, $E_1A_1B_1$, $E_1P_1Q_1$, $P_1$, $E_1Q_1$ and $Q_1$, respectively, and then apply an inclusion operation. Conversely, to transform the complex for the random reaction to the complex for the ordered reaction, apply nonadjacent-vertex identifications so that $\{E_i\}_{i=1}^4$, $\{A_i\}_{i=1}^4$, $\{E_iA_i\}_{i=1}^4$, $\{B_i\}_{i=1}^4$, $\{E_iA_iB_i\}_{i=1}^4$, $\{E_iP_iQ_i\}_{i=1}^4$, $\{P_i\}_{i=1}^4$, $\{E_iQ_i\}_{i=1}^4$ and $\{Q_i\}_{i=1}^4$ are identified with $E$, $A$, $EA$, $B$, $EAB$, $EPQ$, $P$, $EQ$ and $Q$, respectively.

Now we demonstrate that the ordered sequential bisubstrate reaction is not equivalent to the ping-pong bisubstrate reaction. We cannot change the 2-simplices with vertex sets $\{E, A, EA\}$ and $\{E, Q, EQ\}$ in each bisubstrate reaction, since they correspond to identical components and interactions. The ping-pong bisubstrate reaction contains a 1-simplex with vertex set $\{EA, E^*P\}$, which is not in the ordered sequential bisubstrate reaction and would require a vertex split operation applied to EA in the ordered reaction. The new vertex from the vertex split operation, labelled $E^*P$, would however be a vertex in several new simplices, in particular the simplex with vertex set $\{E, EA, E^*P\}$, which is not in the ping-pong reaction. Therefore, we cannot transform the ordered sequential bisubstrate reaction into the ping-pong bisubstrate reaction, so they are not equivalent.

**Morphogen 1**

① Morphogen 1

② Diffusion 1

③ Degradation 1

④ Production 1

⑤ Basal production 1

⑥ Influx 1

⑦ Outflux 1

⑧ Morphogen 1 bound

**Morphogen 2**

⑨ Morphogen 2

⑩ Diffusion 2

⑪ Degradation 2

⑫ Basal production 2

⑬ Influx 2

**Morphogen 3**

⑭ Morphogen 3

⑮ Diffusion 3

⑯ Degradation 3

⑰ Production 3

**Substrate 1**

⑱ Substrate 1

⑲ Diffusion of Substrate 1

⑳ Degradation of Substrate 1

㉑ Basal production of Substrate 1

**Modulator 1**

㉒ Modulator 1

㉓ Diffusion of Modulator 1

㉔ Degradation of Modulator 1

㉕ Production of Modulator 1

**Reactions and Interactions**

㉖ Annihilation between Morphogens 1 and 2

㉗ Self-activation of Morphogen 1

㉘ Activation of Morphogen 2 by Morphogen 1

㉙ Inhibition of Morphogen 1 by Morphogen 2

㉚ Inhibition of Morphogen 1 by Morphogen 3

㉛ Inhibition of Morphogen 3 by Morphogen 1

㉜ Inhibition of an inhibition by Morphogen 2

㉝ Production of Morphogen 2 by Morphogen 1

㉞ Depletion of Substrate 1 by Morphogen 1

㉟ Modulation of Diffusion 1 by Modulator 1

㊱ Modulation of Degradation 1 by Modulator 1

㊲ Inhibition of Modulator 1 by Morphogen 1

㊳ Adsorption of Morphogen 1

㊴ Desorption of Morphogen 1 bound

**Morphogen 1 gradient**

㊵ Monotonic gradient

㊶ Oscillatory gradient

㊷ Local scale-invariance

㊸ Global scale-invariance

**Figure 3.** Ordered set of components for the four Turing-pattern and five positional-information models.

## 3.2. Comparison of Turing-pattern and positional-information models

Here, we apply our methods for model comparison to the two main categories of models for developmental pattern formation, namely Turing-pattern models and positional-information models. We consider the patterning to occur throughout a two-dimensional rectangular domain, where the boundary conditions are always zero-flux on the two opposite sides parallel to the morphogen concentration gradient. We assume that the velocities of the cytoplasm and the growing tissue are negligible, and we therefore assume no advection.

To represent the models as simplicial complexes, we first determine the set of components $\mathcal{C}$ on which the models under consideration are based. We consider five positional-information models and four Turing-pattern models. In this case the general components are: agents, namely morphogens, modulators, and substrates; reactions involving the agents, such as self-activation, activation, inhibition and annihilation; agent degradation; influx and outflux boundary conditions; agent diffusion; profile of the morphogen gradient; and scale invariance of the morphogen gradient. The ordered set of components is shown in figure 3. Note that the components can be given in any order since our definition of distance is invariant with respect to changes in the flat filtration (see lemma 2.17). The distance between models is dependent on the level of conceptual detail used to represent the models, so the distances obtained must be considered with respect to the representative simplicial complexes. The most straightforward way to obtain meaningful comparisons is to use a consistent level of conceptual detail for all models. Here, we take the reference complex as the simplex spanned by the complete set of components for all nine models.

To construct the simplicial representation of each Turing-pattern and positional-information model, we first specify the 0-simplices, which represent the model components, and the 1-simplices, which represent direct interconnections between the components. While the 0- and 1-simplices are specified by the model, to give a combinatorial graph, the higher-dimensional simplices are obtained by forming cliques [36], where possible, incrementally in dimensions 2 and higher. These higher-dimensional simplices indicate higher-dimensional interactions between the corresponding model components.

To illustrate the barcode representation, we use a particular flat filtration called the shortlex filtration, which we first describe. Note that while all flat filtrations can be extended (see proposition 2.13), an extension of the shortlex filtration is not necessarily another shortlex filtration. We begin by applying

the shortlex order to the simplices of the simplicial complex with respect to the assigned vertex order, where the shortlex order is defined as follows ([37], ch. 0, p. 14):

**Definition 3.1 (Shortlex order).** Let $\mathcal{A}$ be a totally-ordered finite set, called the *alphabet*, and let $\mathcal{W}$ be the set of all *words* that are finite sequences of symbols from $\mathcal{A}$. The *shortlex order* on $\mathcal{W}$ orders the words as follows:

(i) Two different words in $\mathcal{W}$ with equal length are ordered according to the alphabetic order of $\mathcal{A}$, therefore lexicographically.
(ii) For two words with unequal lengths, the shorter word precedes the longer word.

Note that, since $\mathcal{A}$ is totally ordered, the shortlex order is also a total order. Applying the shortlex order to a collection of simplices, we first order the simplices by increasing dimension, and then the simplices of the same dimension are ordered lexicographically. Since the set of vertices is totally ordered as specified by Ord, so is the shortlex order for the simplices.

We now define the *shortlex filtration*:

**Definition 3.2 (Shortlex filtration).** Let $\mathcal{C}$ be the set of all components that appear in the collection of models under consideration for comparison, and let the reference complex $\mathcal{R}^{(m)}$ have the shortlex ordering. Define the bijective weight function $w : \mathcal{R}^{(m)} \rightarrow [|\mathcal{R}^{(m)}|]$ by $w(\sigma) = j$, where $j$ is the index of $\sigma$ in the shortlex ordering for $\mathcal{R}^{(m)}$. The *shortlex filtration* of $\mathcal{R}^{(m)}$ is the nested sequence of subcomplexes as defined in definition 2.5.

With this filtration it is possible to define the distances of the models that we have described under the umbrella of the reference complex $\mathcal{R}^{(m)}$. Note that, since our definition of distance is invariant with respect to changes in the flat filtration (see lemma 2.17), the components indicated in figure 3 can be given an arbitrary ordering, resulting in an alternative shortlex filtration.

Here, we describe one positional-information model and one Turing-pattern model, and provide the descriptions of the four additional positional-information models and three additional Turing-pattern models in the electronic supplementary material document. We begin by describing the positional-information annihilation model.

### 3.2.1. Positional-information annihilation model

The annihilation model, and also the opposing gradients model (electronic supplementary material), are mechanisms whereby two opposing morphogen gradients provide size information for developmental patterning. In the annihilation model, the target gene responds to the concentration of Morphogen 1, to which Morphogen 2 irreversibly binds and thereby inhibits the action of Morphogen 1 on activity of transcription, so that the gradient of Morphogen 2 provides size information to the concentration field of Morphogen 1 [38].

The sources of each morphogen are at opposite ends of the domain, and the morphogens interact by an annihilation reaction with rate $k$ that results in global scale-invariant patterning [38,39]. Mathematically, the two morphogen gradients with concentrations $m(\mathbf{x}, t)$ and $c(\mathbf{x}, t)$ can be modelled as

$$\frac{\partial m}{\partial t} = D_m \nabla^2 m - k_m m - kmc \tag{3.1}$$

and

$$\frac{\partial c}{\partial t} = D_c \nabla^2 c - k_c c - kmc \tag{3.2}$$

where $D_m$ and $D_c$ are diffusivities, and $k_m$ and $k_c$ are degradation rates. The boundary conditions for each morphogen are Neumann at both boundaries, with influx at the source and zero flux at the opposite boundary.

For the simplicial representation of the annihilation model, the vertices and corresponding model components are:

— $v_1 \longleftrightarrow$ Morphogen 1
— $v_2 \longleftrightarrow$ Diffusion 1
— $v_3 \longleftrightarrow$ Degradation 1
— $v_6 \longleftrightarrow$ Influx 1
— $v_9 \longleftrightarrow$ Morphogen 2
— $v_{10} \longleftrightarrow$ Diffusion 2

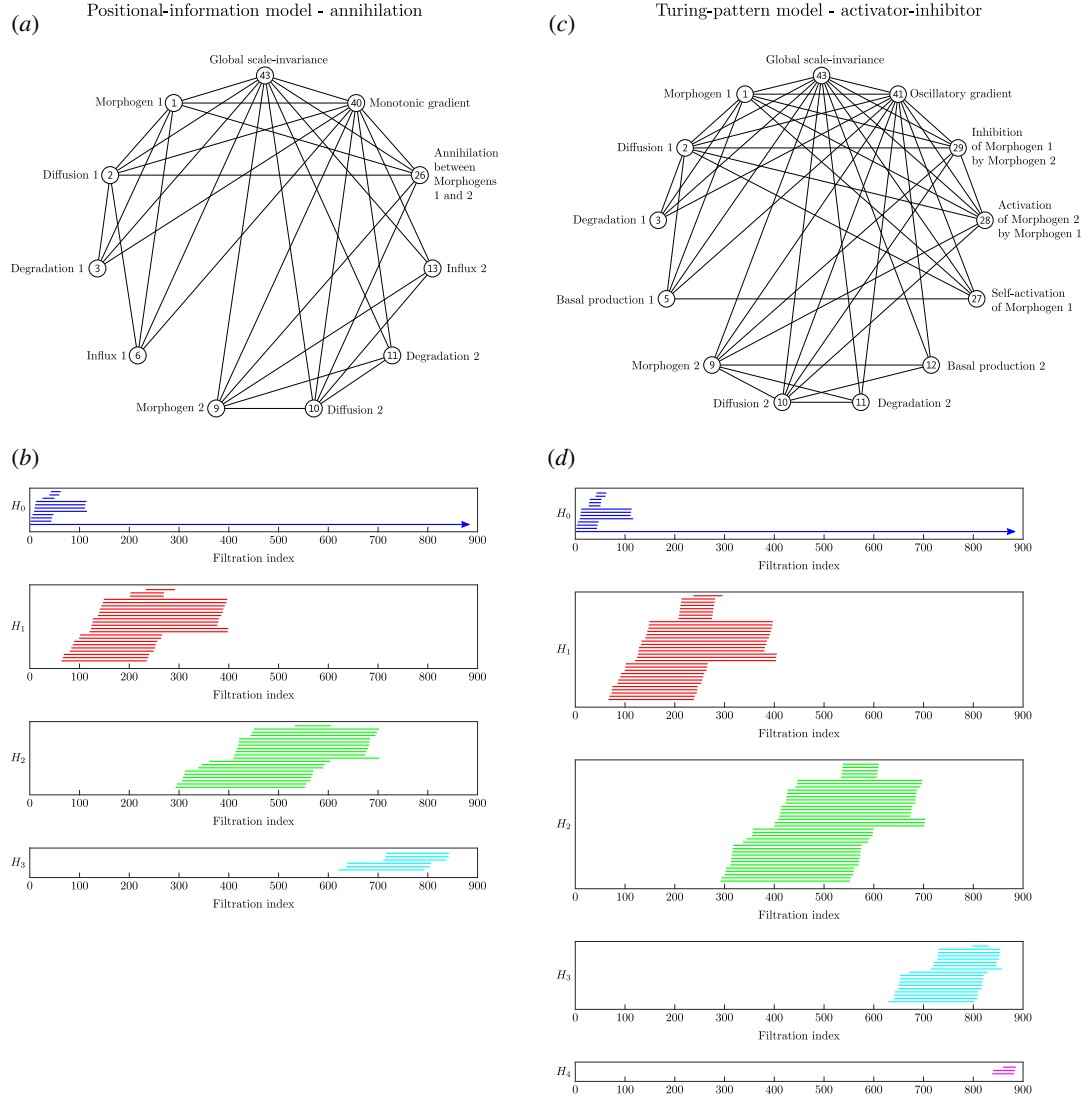

**Figure 4.** Simplicial-complex representations and corresponding persistence barcodes. Positional-information annihilation model: (*a*) 1-skeleton of the four-dimensional simplicial representation; (*b*) persistence barcode. Turing-pattern activator–inhibitor model: (*c*) 1-skeleton of the five-dimensional simplicial representation; (*d*) persistence barcode. Note that the $H_n$ labels on the persistence barcodes refer to the $n$-dimensional homology classes, and the arrows at the right endpoints of certain persistence intervals indicate that the right endpoints are $+\infty$, corresponding to homology classes that are not annihilated in the filtration.

— $v_{11} \longleftrightarrow$ Degradation 2
— $v_{13} \longleftrightarrow$ Influx 2
— $v_{26} \longleftrightarrow$ Annihilation between Morphogens 1 and 2
— $v_{40} \longleftrightarrow$ Monotonic gradient
— $v_{43} \longleftrightarrow$ Global scale-invariance

For the annihilation model, the simplicial complex and corresponding persistence barcode is shown in figure 4*a*,*b*. Note that the simplicial complex is four-dimensional, however, we only show the 1-skeleton of the simplicial complex for simplicity.

We now describe the Turing-pattern activator–inhibitor model.

### 3.2.2. Turing-pattern activator-inhibitor model

The activator–inhibitor system [8,40,41] consists of two diffusible morphogens, an autocatalytic activator with concentration $m(\mathbf{x}, t)$ and a rapidly diffusing inhibitor with concentration $c(\mathbf{x}, t)$. This model can be

represented mathematically as

$$\frac{\partial m}{\partial t} = D_m \nabla^2 m + \frac{\rho m^2}{c(1 + \mu_m m^2)} - k_m m + \rho_m \tag{3.3}$$

and

$$\frac{\partial c}{\partial t} = D_c \nabla^2 c + \rho m^2 - k_c c + \rho_c, \tag{3.4}$$

where $D_m$ and $D_c$ are diffusivities, $\rho_m$ and $\rho_c$ are basal production rates, $k_m$ and $k_c$ are degradation rates, and $\mu_m$ is a saturation constant. The parameter $\rho$ is the *source density*, which measures the general ability of the cells to perform the autocatalytic reaction. The patterning arises through local self-enhancement of the activator, activation of the inhibitor, and long-range inhibition of the activator. We assume that the boundary conditions are zero flux at both boundaries.

For the simplicial representation of the activator–inhibitor model, the vertices and corresponding model components are:

— $v_1 \longleftrightarrow$ Morphogen 1
— $v_2 \longleftrightarrow$ Diffusion 1
— $v_3 \longleftrightarrow$ Degradation 1
— $v_5 \longleftrightarrow$ Basal production 1
— $v_9 \longleftrightarrow$ Morphogen 2
— $v_{10} \longleftrightarrow$ Diffusion 2
— $v_{11} \longleftrightarrow$ Degradation 2
— $v_{12} \longleftrightarrow$ Basal production 2
— $v_{27} \longleftrightarrow$ Self-activation of Morphogen 1
— $v_{28} \longleftrightarrow$ Activation of Morphogen 2 by Morphogen 1
— $v_{29} \longleftrightarrow$ Inhibition of Morphogen 1 by Morphogen 2
— $v_{41} \longleftrightarrow$ Oscillatory gradient
— $v_{43} \longleftrightarrow$ Global scale-invariance

For the activator–inhibitor model, the simplicial complex and corresponding persistence barcode is shown in figure 4c,d. Note that the simplicial complex is five-dimensional, however, we only show the 1-skeleton of the simplicial complex for simplicity.

### 3.2.3. Distances between models

We now apply theorem 2.18 to calculate the distances between the simplicial representations of the five positional-information and four Turing-pattern models. These distances, shown in table 2, indicate the total number of labelled simplices that must be added and subtracted to transform one labelled simplicial complex associated with a model to the labelled simplicial complex of another model.

In table 3, we also show the number of simplices in each simplicial representation of the four Turing-pattern (TP) and five positional-information (PI) models.

We observe that the distance between two models can be relatively large even when the models have similar numbers of simplices of each dimension. For example, the Turing-pattern inhibition-of-an-inhibition model (TP 3) and the Turing-pattern modulation model (TP 4) have similar numbers of simplices in each dimension, however they have the largest distance among the four Turing-pattern models that we compare. This illustrates that the metric accounts for not just the numbers of simplices but also for the labelling of the simplices.

Furthermore, our measure of distance between models is able to reveal when two models are relatively similar, for example the Turing-pattern modulation model (TP 4) is a modification of the Turing-pattern activator–inhibitor (TP 1) model, and these two models have the smallest distance among the four Turing-pattern models that we compare.

### 3.2.4. Equivalence of models

We now apply our notion of model equivalence to compare the positional-information annihilation model with the Turing-pattern activator–inhibitor model. The 1-skeletons of the simplicial representations of the annihilation model and the activator–inhibitor model are shown in figure 4a,c,

**Table 2.** Distances between the four Turing-pattern (TP) and five positional-information (PI) models.

| | PI 2 | PI 3 | PI 4 | PI 5 | TP 1 | TP 2 | TP 3 | TP 4 |
|---|---|---|---|---|---|---|---|---|
| PI 1 (linear gradient) | 40 | 108 | 104 | 112 | 240 | 176 | 376 | 340 |
| PI 2 (synthesis-diffusion-degradation) | | 68 | 96 | 104 | 216 | 152 | 352 | 316 |
| PI 3 (opposing gradients) | | | 120 | 172 | 270 | 220 | 406 | 370 |
| PI 4 (annihilation) | | | | 152 | 268 | 232 | 404 | 368 |
| PI 5 (active modulation) | | | | | 304 | 240 | 440 | 404 |
| TP 1 (activator-inhibitor) | | | | | | 192 | 432 | 100 |
| TP 2 (substrate depletion) | | | | | | | 424 | 292 |
| TP 3 (inhibition of an inhibition) | | | | | | | | 532 |
| TP 4 (modulation) | | | | | | | | |

**Table 3.** Number of $n$-dimensional simplices in the labelled simplicial representations of the four Turing-pattern (TP) and five positional-information (PI) models.

| $n$ | 0 | 1 | 2 | 3 | 4 | 5 |
|---|---|---|---|---|---|---|
| PI 1 (linear gradient) | 6 | 14 | 16 | 9 | 2 | 0 |
| PI 2 (synthesis-diffusion-degradation) | 5 | 9 | 7 | 2 | 0 | 0 |
| PI 3 (opposing gradients) | 10 | 27 | 32 | 18 | 4 | 0 |
| PI 4 (annihilation) | 11 | 33 | 43 | 26 | 6 | 0 |
| PI 5 (active modulation) | 13 | 39 | 47 | 24 | 4 | 0 |
| TP 1 (activator-inhibitor) | 13 | 45 | 70 | 55 | 21 | 3 |
| TP 2 (substrate depletion) | 12 | 37 | 50 | 33 | 10 | 1 |
| TP 3 (inhibition of an inhibition) | 17 | 70 | 119 | 96 | 36 | 5 |
| TP 4 (modulation) | 16 | 65 | 108 | 84 | 30 | 4 |

respectively. We shall demonstrate the equivalence of the two models by applying Operations 1, 3 and 5 to the Turing-pattern activator–inhibitor model to obtain the positional-information annihilation model. Note that, since the three operations are invertible, we could similarly apply operations to the positional-information annihilation model to obtain the Turing-pattern activator–inhibitor model. In accordance with the admissibility requirements for the operations, we regard as equivalent the subsets of model components indicated in figure 5. We consider 'Influx 1' to be equivalent to the combination of 'Basal production 1' and 'Self-activation of Morphogen 1' since they are all a source of 'Morphogen 1'. Similarly, 'Influx 2' is equivalent to 'Basal production 2' since they are both a source of 'Morphogen 2'. Since we are interested in the existence of the pattern-forming gradient and not the particular profile of the gradient, we consider 'Monotonic gradient' and 'Oscillatory gradient' as equivalent. Finally, 'Annihilation between Morphogens 1 and 2', which reduces the concentrations of both morphogens, is equivalent to the combination of 'Activation of Morphogen 2 by Morphogen 1' and 'Inhibition of Morphogen 1 by Morphogen 2'. Indeed, 'Activation of Morphogen 2 by Morphogen 1' and 'Inhibition of Morphogen 1 by Morphogen 2' form an inhibitory cycle whereby 'Morphogen 2' inhibits 'Morphogen 1', which in turn reduces the activation of 'Morphogen 2' by 'Morphogen 1'. Note that these observed equivalences between model components do not suggest that equivalent components are the same, but rather that equivalent components have similar roles in the conceptual frameworks of the corresponding models, which is demonstrated formally through application of the admissible operations on simplicial complexes to ascertain that the models are in the same equivalence class.

We now show that the positional-information annihilation model and the Turing-pattern activator–inhibitor model are in the same equivalence class. While we describe the application of the operations

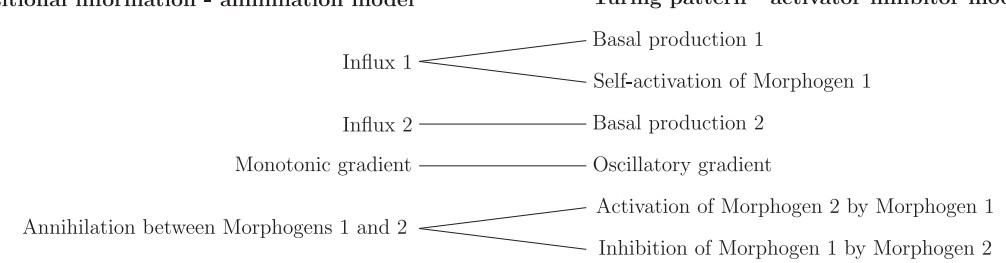

**Figure 5.** Model components considered as equivalent.

on vertices, there are also changes to the higher-dimensional simplices in the simplicial complex. Note that it is easy to automate the process on computer to transform one simplicial complex to another by application of the operations on simplicial complexes. First consider transforming the complex for the Turing-pattern activator–inhibitor model to the complex for the positional-information annihilation model. Perform an adjacent-vertex identification of the two vertices 'Basal production 1' and 'Self-activation of Morphogen 1' to obtain the new vertex 'Influx 1'. The vertex 'Basal production 2' is substituted with the vertex 'Influx 2'. Similarly, we substitute the vertex 'Monotonic gradient' for the vertex 'Oscillatory gradient'. Finally, we perform an adjacent-vertex identification of the two vertices 'Activation of Morphogen 2 by Morphogen 1' and 'Inhibition of Morphogen 1 by Morphogen 2' to obtain the vertex 'Annihilation between Morphogens 1 and 2'. The resulting simplicial complex is that which represents the positional-information annihilation model.

Conversely, to transform the complex for the positional-information annihilation model to the complex for the Turing-pattern activator–inhibitor model we apply the inverse operations. First perform a vertex split of the vertex 'Annihilation between Morphogens 1 and 2' to obtain the two vertices 'Activation of Morphogen 2 by Morphogen 1' and 'Inhibition of Morphogen 1 by Morphogen 2'. Substitute the vertex 'Oscillatory gradient' for the vertex 'Monotonic gradient', and then substitute the vertex 'Basal production 2' for the vertex 'Influx 2'. Finally, apply a vertex split of the vertex 'Influx 1' to obtain the two vertices 'Basal production 1' and 'Self-activation of Morphogen 1'. The resulting simplicial complex is that which represents the Turing-pattern activator–inhibitor model. Since we have only used admissible and invertible operations to transform the simplicial representation of the Turing-pattern model to the simplicial representation of the positional-information model, the two models are in the same equivalence class, and therefore are equivalent. We can also compare the persistence barcodes shown in figure 4b,d, which have a high degree of similarity in accordance with the established equivalence between the positional-information annihilation model and the Turing-pattern activator–inhibitor model.

This analysis demonstrates that the Turing-pattern activator–inhibitor model and the positional-information annihilation model, each one of the main models for their patterning mechanism in developmental biology, are in fact closely related in terms of structure and mechanism. This is not obvious given the nature of the proposed models (e.g. [42,43]), has not previously been reported, and is unexpected given the frequent historical acrimony between proponents of the two mechanisms [30]. The main difference between the positional-information model and the Turing-pattern model is the source of the gradient-forming morphogen, whereby the positional-information mechanism takes advantage of an influx of morphogen from outside the patterning domain, whereas for the Turing-pattern mechanism the morphogen is produced uniformly within the domain. The morphogen influx naturally produces a morphogen gradient in the positional-information mechanism, so there is no requirement for specialized dynamics. In contrast, the Turing-pattern mechanism requires a Turing instability to generate an oscillating gradient from the spatially uniform production of the morphogen, requiring very particular conditions for existence.

It is important to note here that very few models would be equivalent with respect to our strict definition of model equivalence. Indeed, with the various model components and their interconnections, along with higher-dimensional interactions, it is unlikely that two simplicial representations of two models could be transformed into each other by application of admissible operations. Our established equivalence of the positional-information annihilation model and the Turing-pattern activator–inhibitor model, with only a few admissible operations required, therefore demonstrates that the two models are conceptually very similar from our considered perspective. This is further substantiated by the ability to

employ adjacent-vertex identification operations, or conversely vertex-split operations, which require identical conceptual interconnections for the model concepts involved.

## 4. Conclusion

In this article, we present a new methodology for comparing models based on the relationships between various model aspects. Compared to some previous attempts at comparing model structures [42] our approach is readily automatable, and thus able to meet the demands of large-scale modelling attempts and model-curation projects [44].

Representing models as labelled simplicial complexes allows us to determine meaningful distances between models, and persistent homology provides an alternative and often simplified representation of the models. In addition to a measure of distance, we have also developed and applied the concept of equivalence to compare model features. Model equivalence, as developed here, gives more nuanced insights into the relationships between different models and allows us to detect non-trivial similarities between mathematical models. Our analysis of the Turing-pattern activator–inhibitor model and the positional-information annihilation model shows that these two models are more similar than had previously been suggested [30]. Given the conditions outlined in the discussion they are in fact equivalent, something that we had not expected and which to our knowledge has not been seen or considered before. This is one example demonstrating the potential insights that can be gained from our formalism for model comparison.

We foresee more need for this in the immediate future: modelling is increasing in importance in the life and biomedical sciences, yet remains to be fallible and often poorly grounded in reality [45,46]. The ability to compare, contrast, reconcile, and triage potentially large sets of models will aid in making mathematical modelling more helpful in biology.

While we have applied our methodology to similar types of models, namely systems of reaction–diffusion equations, another important aspect of our formalism is that we can compare different models, irrespective of how distinct they are in form and size: by identifying the model components and their interconnections we can represent any model as a simplicial complex which allows for direct comparison with any other simplicial representation of a model. Identifying differences, but also highlighting similarities in model structures, can aid our understanding of scientific problems. For example, in the particular example of Turing-pattern versus positional-information mechanisms, this approach can really resolve long-standing problems and trigger the search for a more synthetic approach [30,42]. Because of its flexibility and rigorous grounding our methodology is universally applicable to all models.

Data accessibility. Data and relevant code (both MATLAB and Julia) for this research work are stored in GitHub: https://github.com/DrSeanTVittadello/ModelComparison2021 and have been archived within the Zenodo repository: https://doi.org/10.5281/zenodo.5510811.

Authors' contributions. S.T.V. and M.P.H.S. conceived and planned this analysis; S.T.V. conducted the research, performed the analysis, and drafted the manuscript; all authors reviewed, edited, and approved the final version.

Competing interests. We declare we have no competing interests.

Funding. The authors gratefully acknowledge funding through a 'Life?' programme grant from the Volkswagen Stiftung. M.P.H.S. is funded through the University of Melbourne Driving Research Momentum program.

Acknowledgements. We thank members of the Theoretical Systems Biology group at the University of Melbourne and Imperial College London, Heike Siebert (FU Berlin), James Briscoe (The Francis Crick Institute) and Mark Isalan (Imperial College London) for helpful discussions on Turing patterns and model discrimination.

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
