## [Peer Review File · Royal Society Open Science]

Review History

Decision letter (RSOS-211361.R0)

Dear Dr Vittadello

On behalf of the Editors, we are pleased to inform you that your Manuscript RSOS-211361 "Model comparison via simplicial complexes and persistent homology" has been accepted for publication in Royal Society Open Science without requiring additional revisions.

Before we are able to pass on the paper to the production team, there are a number of minor matters that we need you to address, please. Specifically:

1. Per the guidance below, please ensure you provide the journal with the original, editable files for the accepted version of your paper.
2. At this stage, we ask that you please archive your GitHub code within the Zenodo repository: <https://guides.github.com/activities/citable-code/>. By doing this, a formal, citable DOI will be

associated with your data record, and an open license (CC-BY preferred) can be applied to your data. We would then ask that you please update your data availability statement to read as:

"Data and relevant code for this research work are stored in GitHub: [GitHub URL here] and have been archived within the Zenodo repository: <https://doi.org/zenodo.....> [ref number].

Please submit your revised manuscript and required files (see below) no later than 7 days from today's (ie 13-Sep-2021) date. Note: the ScholarOne system will 'lock' if submission of the revision is attempted 7 or more days after the deadline. If you do not think you will be able to meet this deadline please contact the editorial office immediately.

on behalf of Dr Jose Carrillo (Associate Editor) and Mark Chaplain (Subject Editor)
openscience@royalsociety.org

===PREPARING YOUR REVISION IN SCHOLARONE===

Author's Response to Decision Letter for (RSOS-211361.R0)

See Appendix A.

Decision letter (RSOS-211361.R1)

Dear Dr Vittadello,

I am pleased to inform you that your manuscript entitled "Model comparison via simplicial complexes and persistent homology" is now accepted for publication in Royal Society Open Science.

on behalf of Dr Jose Carrillo (Associate Editor) and Mark Chaplain (Subject Editor)
openscience@royalsociety.org

Appendix A

As requested by the Editor, we have attended to the following:

- 1) We have provided Royal Society Open Science with the original and editable LaTeX files for the accepted version of our manuscript.
- 2) We have archived our GitHub code within the Zenodo repository, and updated our data availability statement accordingly.